# Cooperative adsorption of carbon disulfide in diamine-appended metal–organic frameworks

C. Michael McGuirk [1], Rebecca L. Siegelman [1,2], Walter S. Drisdell[3], Tomče Runčevski[1,2], Phillip J. Milner, Julia Oktawiec, Liwen F. Wan[4], Gregory M. Su [5], Henry Z.H. Jiang, Douglas A. Reed, Miguel I. Gonzalez, David Prendergast [4] & Jeffrey R. Long [6]

Over one million tons of $CS_2$ are produced annually, and emissions of this volatile and toxic liquid, known to generate acid rain, remain poorly controlled. As such, materials capable of reversibly capturing this commodity chemical in an energy-efficient manner are of interest. Recently, we detailed diamine-appended metal–organic frameworks capable of selectively capturing $CO_2$ through a cooperative insertion mechanism that promotes efficient adsorption–desorption cycling. We therefore sought to explore the ability of these materials to capture $CS_2$ through a similar mechanism. Employing crystallography, spectroscopy, and gas adsorption analysis, we demonstrate that $CS_2$ is indeed cooperatively adsorbed in *N,N*-dimethylethylenediamine-appended $M_2$(dobpdc) (M = Mg, Mn, Zn; dobpdc$^{4-}$ = 4,4′-dioxi-dobiphenyl-3,3′-dicarboxylate), via the formation of electrostatically paired ammonium dithiocarbamate chains. In the weakly thiophilic Mg congener, chemisorption is cleanly reversible with mild thermal input. This work demonstrates that the cooperative insertion mechanism can be generalized to other high-impact target molecules.

[1] Department of Chemistry, University of California, Berkeley, California 94720, USA. [2] Materials Sciences Division, Lawrence Berkeley National Laboratory, Berkeley, California 94720, USA. [3] Chemical Sciences Division, Lawrence Berkeley National Laboratory, Berkeley, California 94720, USA. [4] The Molecular Foundry, Lawrence Berkeley National Laboratory, Berkeley, California 94720, USA. [5] Advanced Light Source, Lawrence Berkeley National Laboratory, Berkeley, California 94720, USA. [6] Department of Chemical and Biomolecular Engineering, University of California, Berkeley, California 94720, USA. Correspondence and requests for materials should be addressed to J.R.L. (email: jrlong@berkeley.edu)

Metal–organic frameworks are modular, crystalline porous materials composed of inorganic nodes bridged by multitopic organic linkers, and have been widely explored for the selective capture and storage of small molecules[1–3]. Adsorption in these materials is most commonly characterized by a Type I isotherm profile, with large and energetically-costly pressure or temperature swings generally required to achieve high cycling capacities (Fig. 1a)[4–7]. Alternatively, flexible and pore-gating adsorbents with Type V adsorption profiles (Fig. 1b) can increase the energy efficiency of cycling through access to large cyclic capacities with small changes in pressure or temperature[8–11]. Additionally, this favorable Type V adsorption profile has recently been reported in metal–organic frameworks that chemically bind small molecules in a cooperative fashion, showing minimal guest uptake below a temperature-dependent threshold pressure, followed by a sudden and sharp rise, or step, in adsorption[12,13].

By appending alkylethylenediamines to the metal sites lining the one-dimensional hexagonal channels of the metal–organic frameworks $M_2$(dobpdc) (M = Mg, Mn, Fe, Co, Zn; dobpdc$^{4-}$ = 4,4′-dioxidobiphenyl-3,3′-dicarboxylate; Fig. 2a), we recently discovered the first example of a material displaying cooperative adsorption via a unique, chain-forming mechanism[12]. In these materials, at a specific threshold pressure $CO_2$ inserts into the metal–amine bonds, a process that is coupled with proton transfer to induce the formation of one-dimensional ammonium carbamate chains running along the corners of the hexagonal channels of the framework. This reactive mechanism imbues diamine-appended $M_2$(dobpdc) with a chemical specificity often not displayed in other materials with Type V adsorption profiles[9,14,15]. The resulting electrostatic, covalent, and dative interactions within these chains can be disrupted with relatively small temperature and/or pressure swings, returning free $CO_2$ and the native diamine-appended framework. In the adsorption isotherm, this mechanism manifests as sharp, temperature dependent adsorption/desorption steps that can be controlled by varying the constituent metal and/or diamine species[12,16]. The chemical tunability and energy efficient cycling possible with these materials have prompted the development of adsorbents tailored specifically for the low-energy capture of $CO_2$ from flue gas streams[16,17].

Despite the practical advantages, thus far all investigations of this cooperative chemical adsorption process have been limited to $CO_2$[12,16,17], and it is therefore of interest to evaluate the applicability of this mechanism to other relevant adsorbates. In determining viable candidate adsorbates, three necessary properties were identified. Considering the mechanism of $CO_2$ uptake, it is clear that other viable adsorbates must (i) possess a dipole or quadrupole moment that promotes insertion into metal–amine bonds, (ii) form a coordinatively stable, Brønsted acidic species upon insertion that is capable of electrostatic pairing, and (iii) form bonds with the amine and metal that are relatively labile to allow reversible adsorption under mild conditions. Specifically, Brønsted acidity upon insertion is critical to allow proton transfer to the proximal amine and the resultant formation of electrostatic pairs, yielding site-to-site communication and a cooperative, chemically specific adsorption process. Given existing precedent for dithiocarbamate formation and direct insertion into metal–amine bonds[18,19], the commodity chemical carbon disulfide ($CS_2$) was identified as a promising candidate to assess the generalizability of this cooperative adsorption mechanism (Fig. 2b).

Over one million tons of $CS_2$ are produced annually for use as a non-polar solvent, a $C_1$ synthon, and, most prodigiously, as a processing reagent in the production of viscose rayon and cellophane[20]. Yet this large-scale production presents a number of hazards, namely the volatility and flammability of $CS_2$, the well-documented links to the production of acid rain, and, not least of all, cardiovascular and neurological disease in factory workers[21–23]. Despite these environmental and biological considerations, the development of porous adsorbents for the capture of $CS_2$ has been nearly unexplored to date[24–26]. We therefore sought to investigate whether diamine-appended $M_2$(dobpdc) framework materials could cooperatively and reversibly adsorb this highly toxic commodity chemical. Herein, we present detailed characterization of $CS_2$ adsorption in the N,N-dimethylethylenediamine-appended forms of $M_2$(dobpdc) (M = Mg, Mn, Zn) via diffraction, spectroscopy, and gas adsorption-based techniques, including crystallographic evidence of a dithiocarbamate coordination mode that is, to our knowledge, without precedent.

## Results

**Synthesis and Characterization.** Microcrystalline powders of $M_2$(dobpdc) (M = Mg, Mn, Zn) were prepared and characterized according to previously published procedures (see Large Scale Synthesis of $M_2$(dobpdc) (M = Mg, Mn, Zn) in Methods section, Supplementary Figures 1–5, 33–34)[16,17]. The methanol-solvated frameworks were grafted with the 1°/3° diamine N,N-dimethylethylenediamine (mm-2), and the extent of the mm-2 loading,

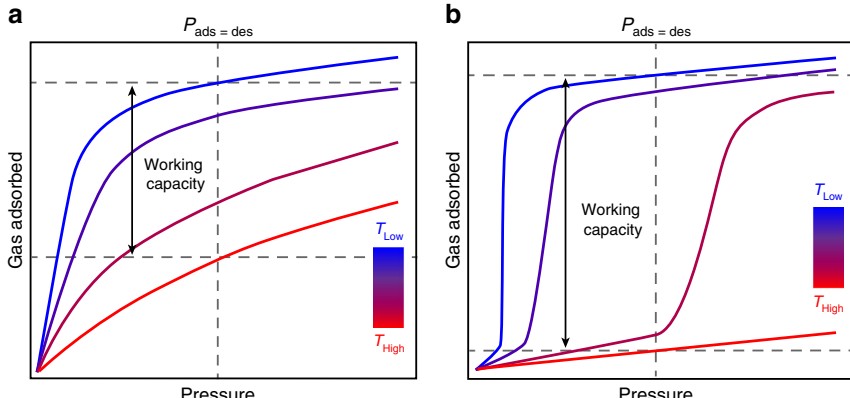

**Fig. 1** Idealized gas adsorption isotherms. Comparison of the temperature-dependent adsorption behavior for a microporous material exhibiting **a** classical Type I Langmuir adsorption and **b** cooperative step-shaped adsorption (Type V). Relative working capacities achieved with both types of porous materials when $P_{ads} = P_{des}$ are indicated by double-headed arrows. Importantly, cooperative adsorption profiles allow for greater working capacities with smaller temperature swings

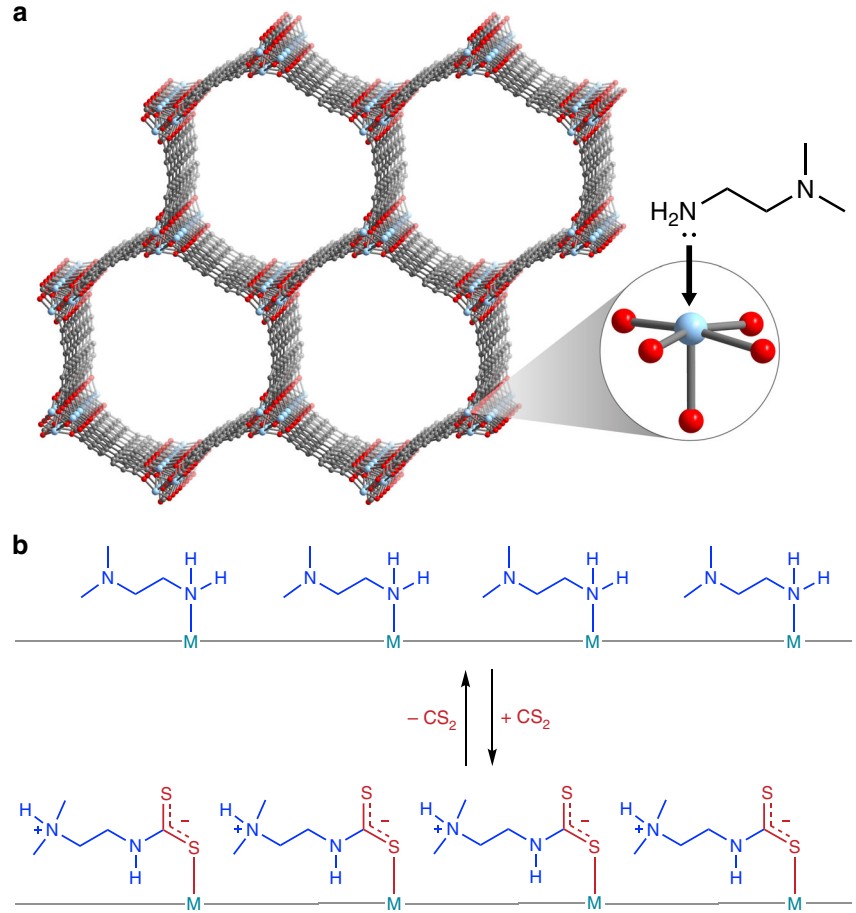

**Fig. 2** Adsorbent and proposed cooperative adsorption process. **a** Representative structure of $M_2$(dobpdc) (dobpdc$^{4-}$ = 4,4'-dioxidobiphenyl-3,3'-dicarboxylate). Light blue, red, and gray spheres represent M, O, and C atoms, respectively; H atoms are omitted for clarity. The inset shows an expanded view of an open metal site appended with the diamine *N,N*-dimethylethylenediamine (mm-2). **b** Proposed reversible process for cooperative $CS_2$ adsorption in mm-2-$M_2$(dobpdc), illustrating insertion of $CS_2$ into the M–N bonds, proton transfer, and formation of electrostatically paired ammonium dithiocarbamate chains

framework stability, and framework porosity were evaluated (Supplementary Figures 6–9, 15–23). The diamine mm-2 was selected due to the preferential binding of the primary amine to the metal, as determined by single-crystal X-ray diffraction[16], which obviates undesirable non-cooperative adsorption pathways due to the inability of the tertiary amine to react with $CS_2$[27].

**Adsorption Studies**. Adsorption of $CS_2$ was first measured in mm-2–$Mg_2$(dobpdc), owing to the high stability and low $CO_2$ insertion pressure of this material relative to its transition metal analogs. At 25 °C, the $CS_2$ adsorption isotherm displays minimal uptake below a threshold pressure of 5 mbar, at which point step-shaped adsorption occurs and approaches saturation (~8 mmol $g^{-1}$) at 25 mbar (Fig. 3a; for a log plot of the isotherm, Supplementary Figure 24; for a plot of $P/P_0$, Supplementary Figure 28). Notably, at 25 °C, mm-2–$Mg_2$(dobpdc) does not adsorb $CO_2$ from air (i.e., 0.4 mbar $CO_2$), thus $CO_2$ should not interfere in a $CS_2$ scavenging process in ambient atmosphere[16]. At elevated temperatures, the adsorption step shifts substantially to higher pressures, with no step occurring below 300 mbar at 120 °C (Fig. 3a; Supplementary Figure 25). Using linear spline interpolation and the Clausius–Clapeyron relationship[28], the differential enthalpy of adsorption ($\Delta h_{ads}$) at a loading of 2 mmol $g^{-1}$ (i.e., corresponding to the step region) was calculated to be −55 ± 5 kJ $mol^{-1}$ (± = standard error; Supplementary Figures 26–27).

When $CS_2$ adsorption was measured for $Mg_2$(dobpdc) appended with isopentylamine, a primary monoamine with a similar steric profile to mm-2 (Supplementary Figures 10–14), similar step-shaped adsorption was not observed and a $\Delta h_{ads}$ of just −26.0 ± 0.3 kJ $mol^{-1}$ was calculated (Supplementary Figures 31–32). This result confirms that the diamine plays a central role in promoting the observed adsorption behavior and excludes condensation as the origin of the step-shaped isotherm. Individual isotherms of various gases at 75 °C, including $CO_2$, suggest chemical specificity for cooperative reactive adsorption of $CS_2$ (Fig. 3b)[16], with a noncompetitive selectivity for $CS_2$ over $H_2O$ at 75 °C mbar and 30 mbar of 11.1[17]. Note, owing to instrument limitations, water isotherms could only be collected to 35 mbar ($P/P_0 = 0.09$)[16].

Isotherm data collected at 25 °C for mm-2–$Mn_2$(dobpdc) and mm-2–$Zn_2$(dobpdc) exhibit similar step-shaped adsorption profiles with slightly lower threshold pressures of 0.5 and 2 mbar, respectively (Fig. 3c; Supplementary Figures 29–30). The lower $CS_2$ threshold pressures are in stark contrast to those observed for $CO_2$ adsorption, in which the lower oxophilicity and higher octahedral metal complex stability of the first-row transition metals contributes to considerably higher step pressures[12,29]. Since thiophilicity trends inversely with oxophilicity[30], the observed threshold pressure relationship of Mn < Zn ≤ Mg for $CS_2$ adsorption implicates the formation of M–S bonds in the adsorption mechanism. The overall similar threshold pressures are an apparent culmination of the competition

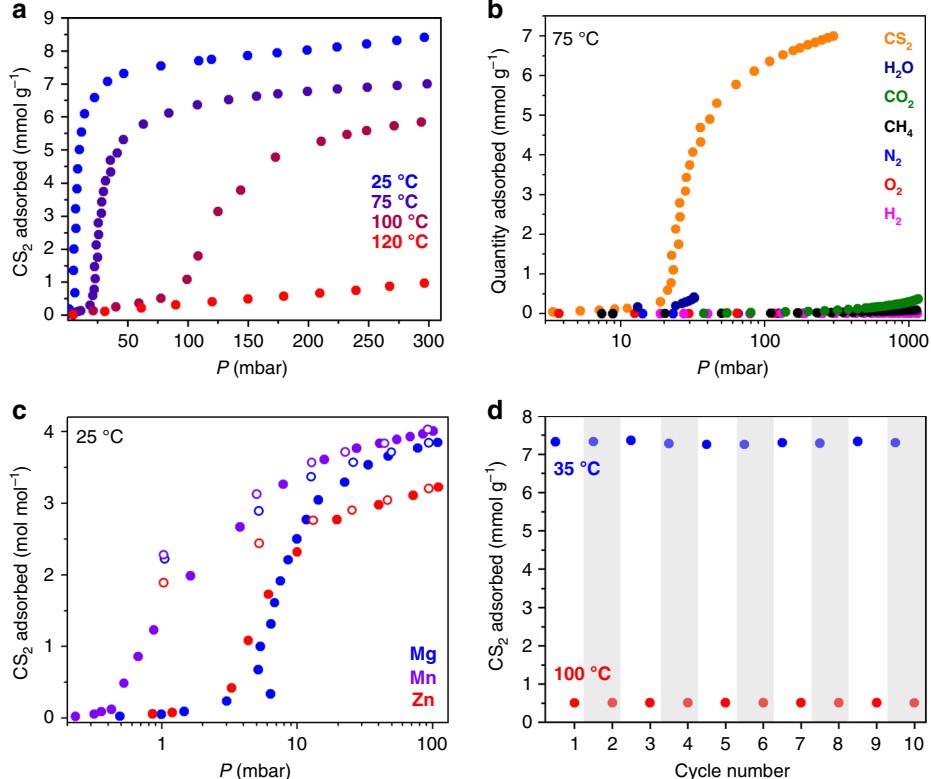

**Fig. 3** CS$_2$ adsorption isotherms and cycling. **a** CS$_2$ adsorption isotherms for mm-2–Mg$_2$(dobpdc) at various temperatures. Adsorption capacity is in units of mmol g$^{-1}$. **b** Comparison of individual adsorption isotherms of various gases at 75 °C collected for mm-2–Mg$_2$(dobpdc) with pressure plotted on a logarithmic scale and adsorption capacity in units of mmol g$^{-1}$. **c** CS$_2$ adsorption isotherms collected for mm-2–M$_2$(dobpdc) (M = Mg, Mn, Zn) at 25 °C with pressure plotted on a logarithmic scale. Adsorption capacity is in units of mol mol$^{-1}$. Filled circles correspond to adsorption and open circles to desorption. **d** Ten temperature-swing adsorption–desorption cycles for mm-2–Mg$_2$(dobpdc) under 75 mbar of CS$_2$. Adsorption occurred at 35 °C (blue circles), and desorption occurred at 100 °C (red circles). Temperature was equilibrated for 30 min between measurements. White and gray columns illustrate successive adsorption–desorption cycles

between octahedral complex stability (Mg < Mn < Zn) and thiophilicity (Mg < Mn < Zn) trends. A pronounced hysteresis is observed for all three frameworks upon desorption at 25 °C (Fig. 3c). At the lower pressure limit of 1 mbar, substantial CS$_2$ remains adsorbed within the materials, coinciding closely to the expected capacities of 4.00, 3.57, and 3.44 mmol g$^{-1}$ (2.0 mol mol$^{-1}$) for one CS$_2$ molecule adsorbed per diamine in the Mg, Mn, and Zn phases, respectively (Supplementary Figure 29). Thus, at 25 °C, the full adsorption capacity of these materials is appreciably higher than 1:1 CS$_2$:diamine. Accordingly, we hypothesized that intrapore condensation occurs along with chemisorption, as will be described in greater detail below. Of note, no similar phenomenon is observed during CO$_2$ adsorption under similar conditions.

Infrared spectroscopy was employed to probe the nature and persistence of the species formed upon CS$_2$ adsorption. Bands centered at 1085–1089 and 953–955 cm$^{-1}$ arising from C–S vibrations of a dithiocarbamate species were observed for all three frameworks (Supplementary Methods; Supplementary Figures 35–37)[19,31] and exhibited no apparent change in intensity after 16 h under dynamic vacuum (~0.13 mbar) at 25 °C. No appreciable difference in band position was observed when CS$_2$ exposure was conducted using a gas adsorption instrument or through vapor exposure in open atmosphere. Moreover, these bands are fully evident in framework pre-saturated with water, supporting that moisture does not preclude this adsorption process (Supplementary Methods; Supplementary Figure 82). In contrast, bands evident of CS$_2$ adsorption in unfunctionalized Mg$_2$(dobpdc) are dramatically reduced upon pre-saturation with water, as expected

for adsorption in an open metal-site framework (Supplementary Figure 83)[32]. While the irreversible binding at ambient temperature is useful for preventing leaching of toxic CS$_2$, regenerable adsorbents are desirable to enable material recyclability. Importantly, after 15 min under reduced pressure (~0.13 mbar) at 100 °C, dithiocarbamate vibrations were completely lost from CS$_2$-exposed mm-2–Mg$_2$(dobpdc) (Supplementary Figure 35), and a similar effect was seen for the Mn framework (Supplementary Figure 36). In contrast, the C–S vibrations for the Zn analog persisted, even upon heating as high as 150 °C (Supplementary Figure 37), signaling irreversible chemisorption through a strong Zn–S bond for the most thiophilic metal in the series[30].

To investigate the integrity of the mm-2-appended Mg and Mn frameworks after treatment with CS$_2$, thermogravimetric analysis (TGA) was used to measure CO$_2$ adsorption isobars for the regenerated samples. Previously, we have shown isobaric CO$_2$ adsorption using TGA to be a reliable and rapid tool for studying cooperative adsorption in these materials[12,16,17,33]. Therefore, if CS$_2$ adsorption is cleanly reversible, the re-activated material would be able to cooperatively adsorb CO$_2$ (Of note, the toxicity of CS$_2$ precludes its use as the adsorbate in this flow-based technique). Following heating of CS$_2$-dosed mm-2–Mg$_2$(dobpdc) above 100 °C under N$_2$ flow to drive off CS$_2$ and subsequent slow cooling under CO$_2$ flow (Supplementary Methods), we observed similar isobaric step-shaped CO$_2$ adsorption to that of a control sample, confirming that mm-2-Mg$_2$(dobpdc) can be rapidly regenerated (Supplementary Figures 38–41). Additionally, mass spectrometry coupled with TGA revealed that intact CS$_2$ is also recovered under these conditions (Supplementary Figures 42–44).

In contrast, mm-2–Mn$_2$(dobpdc) showed close to no CO$_2$ uptake under the same conditions, despite the observed loss of C–S FT–IR vibrations and significant mass loss at elevated temperatures (Supplementary Figures 45–46), signaling that an alternative deleterious pathway occurs during thermolysis. Although this pathway has not been extensively characterized, thermal decomposition of transition metal–dithiocarbamate complexes is known to alternatively produce H$_2$S, thiocyanates, and metal sulfides[31]. Although cooperative CO$_2$ adsorption is readily reversible in all studied diamine–M$_2$(dobpdc) systems[12,16], only the minimally thiophilic Mg framework, with the weakest apparent M–S bond, is capable of reversible CS$_2$ adsorption[30]. Therefore, while Zn and Mn congeners can be used to scavenge lower pressures of CS$_2$, mm-2–Mg$_2$(dobpdc) is superior where adsorbent recyclability is desired.

The step-shaped adsorption profile, thermal sensitivity, and reversibility of CS$_2$ adsorption in mm-2–Mg$_2$(dobpdc) promotes a usable capacity of ~6.8 mmol g$^{-1}$ (52 wt %, 1.7 mol amine$^{-1}$) with a 65 °C temperature swing between 35 and 100 °C under 75 mbar of CS$_2$. Under 300 mbar of CS$_2$, a temperature swing between 25 and 120 °C yields a calculated usable capacity of 7.4 mmol g$^{-1}$. This represents release of almost the entire adsorption capacity under these conditions with mild thermal input. Following ten adsorption–desorption cycles (Supplementary Methods; adsorption at 35 °C and 75 mbar, and desorption at 100 °C and 75 mbar; Fig. 3d), no apparent change was observed in the capacity or integrity of mm-2–Mg$_2$(dobpdc), as confirmed by $^1$H NMR spectroscopy and powder X-ray diffraction (Supplementary Figures 47–48). As a representation of a process in which CS$_2$ is desorbed under a higher pressure of CS$_2$, a working capacity of ~6.3 mmol g$^{-1}$ was calculated for adsorption at 35 °C under 75 mbar CS$_2$ and desorption at 120 °C under 300 mbar CS$_2$ (the highest measurable pressure herein). The adsorption behavior of mm-2–Mg$_2$(dobpdc) affords advantages over recently-reported ion-exchanged zeolites, which exhibit a CS$_2$ capacity of ~0.6 mmol g$^{-1}$ at 25 °C in breakthrough experiments with N$_2$ as the carrier gas under atmospheric pressure, and require regeneration at 400 °C[34].

**X-ray Diffraction Studies.** To elucidate the origin of the observed step-shaped CS$_2$ adsorption profiles, we turned to X-ray diffraction. Single crystals of mm-2–Zn$_2$(dobpdc) exposed to CS$_2$ vapor ex situ undergo a single-crystal-to-single-crystal transformation in which CS$_2$ inserts into the Zn–N bonds (Fig. 4; Supplementary Methods; Supplementary Table 1). Proton transfer from the metal-bound primary amine to the tertiary amine of a neighboring mm-2 group generates an ion-paired ammonium species that propagates down the framework channels as extended ammonium dithiocarbamate chains, the apparent source of the observed step-shaped isotherms. Although the vast majority of dithiocarbamates form strong-field bidentate ligands upon CS$_2$ insertion into metal–amine bonds in molecular complexes[19,31], the robust coordination geometry of Zn$_2$(dobpdc) prevents further ligand displacement, notably forcing the dithiocarbamate into a hydrogen-bonded $\kappa^1$ mode, which is to the best of our knowledge a crystallographically unprecedented binding motif, exhibiting the longest reported Zn–S bond distance (2.4830(12) Å) for a monodentate dithiocarbamate[19,35–38]. This bond length strongly suggests that the anionic character of the dithiocarbamate is considerably delocalized across the moiety, rather than primarily localized on the metal-bound sulfur, in stark contrast to monodentate molecular analogs[35–37].

Two coexisting ammonium dithiocarbamate conformations were observed (Fig. 4b, c). In the major conformation (Fig. 4b), the dithiocarbamate proton forms a hydrogen bond with a bridging phenoxide oxygen atom of the ligand (N···O = 3.004 (10) Å). In the minor conformation (Fig. 4c), the dithiocarbamate proton participates in a stronger hydrogen-bonding interaction with the non-bridging carboxylate oxygen atom of the ligand (N···O = 2.815(9) Å), which has been identified as a preferential secondary binding site in previous crystallographic studies[16,17,39]. Despite this additional stabilization, adjacent chains of this mode appear to overlap in the $ab$ plane (Fig. 4c, top). This physical impossibility implies that neighboring chains in the $ab$ plane exhibit different or alternating conformations, with possible alternation of conformations down the $c$ axis as

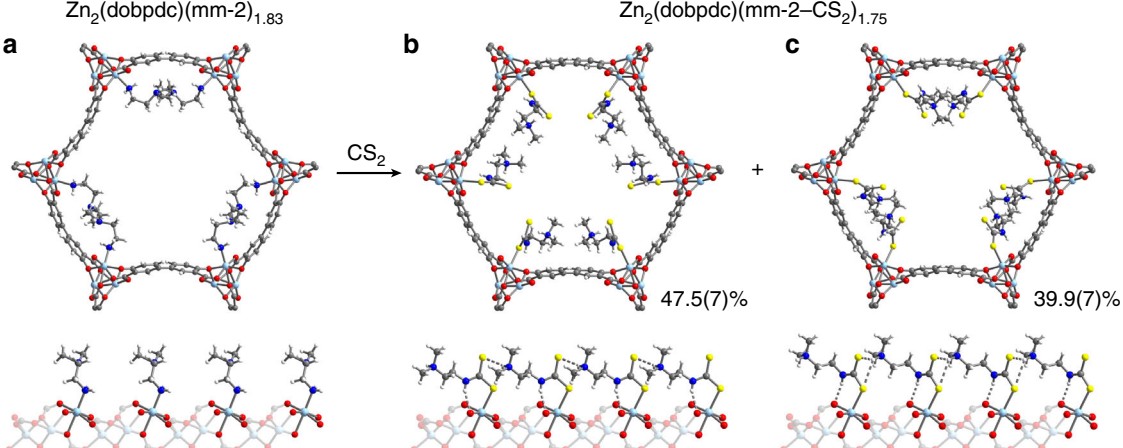

**Fig. 4** Single-crystal X-ray diffraction structures. Portions of the mm-2–Zn$_2$(dobpdc) and CS$_2$-inserted mm-2–Zn$_2$(dobpdc) structures as viewed perpendicular to the $ab$ plane (top) and along the $c$ axis (bottom). All structures were collected at 100 K and refined as inversion twins in either space group $P3_121$ or $P3_221$. For clarity, all images are shown in the $P3_221$ space group. **a** The structure of mm-2–Zn$_2$(dobpdc) showing diamines coordinated to the Zn(II) centers. **b**, **c** Two coexisting ammonium dithiocarbamate chain conformations observed for CS$_2$-inserted mm-2–Zn$_2$(dobpdc). The asymmetric unit consists of a single metal, half of the ligand, and the partial occupancy ammonium dithiocarbamate units in **b** and **c**, which branch from the same metal-bound sulfur atom. Apparent overlap of adjacent chains in the $ab$ plane in **c** implies that neighboring chains exhibit different or alternating conformations, with alternation also possible down the $c$ axis. (Adjacent chains of different conformations do not overlap.) The 10% of Zn(II) sites without dithiocarbamate contain a mixture of solvent, water, CS$_2$, or unreacted diamine that could not be modeled. Light blue, blue, red, gray, yellow, and white spheres represent Zn, N, O, C, S, and H atoms, respectively

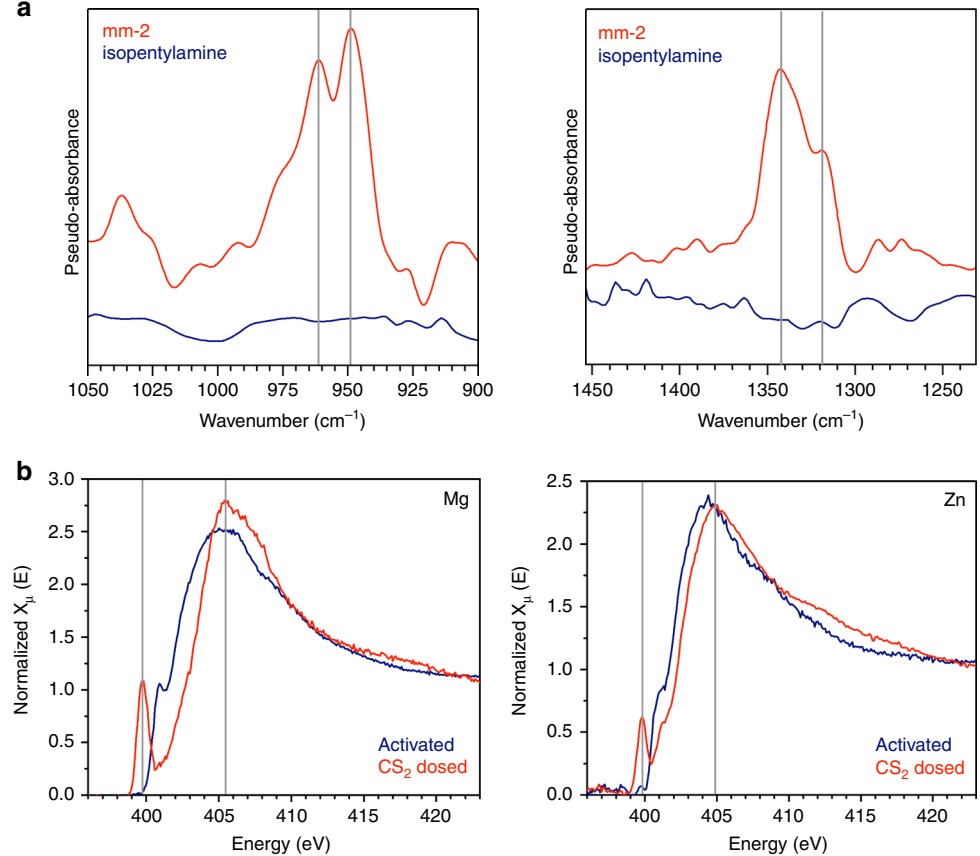

**Fig. 5** Spectroscopic characterization of CS$_2$ Adsorption. **a** In situ diffuse reflectance FT–IR spectra collected upon exposure of activated mm-2–Mg$_2$(dobpdc) and isopentylamine–Mg$_2$(dobpdc) to the vapor pressure of CS$_2$ at 25 °C. Effects from physisorbed CS$_2$ on the framework have been subtracted as the baseline to generate difference plots. Two different sets of distinct bands corresponding to new C–S (left) and C–N (right) vibrations are observed for mm-2–Mg$_2$(dobpdc) (red traces), but not isopentylamine–Mg$_2$(dobpdc) (blue traces). This observation is in agreement with the presence of two distinct ammonium dithiocarbamate chain conformations. **b** In situ NEXAFS spectra collected at the N K-edge upon exposure of activated mm-2–Mg$_2$(dobpdc) (left) and activated mm-2–Zn$_2$(dobpdc) (right) to the vapor pressure of CS$_2$ at 25 °C. After exposing the activated samples to CS$_2$, the environmental cell was evacuated to remove physisorbed CS$_2$ before the measurement was taken. Both congeners show a new pre-edge feature and a shift of the main edge upon CS$_2$ exposure, indicative of ammonium dithiocarbamate formation. All major spectral changes have been computationally reproduced (Supplementary Figures 68–71)

well. However, owing to the averaging nature of SCXRD and preservation of symmetry upon CS$_2$ adsorption, the relative orientation of each conformation cannot be determined definitively directly from the diffraction data. Space-filling models of the two chain conformations (Supplementary Figure 49) demonstrate that significant porosity remains after CS$_2$ insertion, providing space for additional physisorption. Intriguingly, the CS$_2$ hydrolysis mechanism of the metalloenzyme CS$_2$ hydrolase isolated from *Acidianus* A1-3 is postulated to occur through a similar monodentate mode upon insertion into a Zn-ligand bond[40].

Because single crystals of the Mg and Mn analogs of the metal–organic framework could not be realized, synchrotron powder X-ray diffraction patterns of polycrystalline powders of mm-2–Mg$_2$(dobpdc) and mm-2–Mn$_2$(dobpdc) exposed to 50 mbar of CS$_2$ at 25 °C were collected at the Advanced Photon Source (Supplementary Methods; Supplementary Figures 50–51). Starting from the Zn single-crystal structural model, Rietveld refinement confirmed ammonium dithiocarbamate chain formation in both frameworks, affording the first crystallographic evidence of a monodentate Mg–dithiocarbamate structure, and only the second such structure for Mn[41]. The single-crystal and powder X-ray diffraction analyses yield structural models that

support an adsorptive process consisting of CS$_2$ insertion, proton transfer, and one-dimensional chain formation for all three studied mm-2–M$_2$(dobpdc) frameworks. Due to the unique reversibility of CS$_2$ adsorption for mm-2–Mg$_2$(dobpdc), this material was selected for further spectroscopic studies to corroborate the crystallographically observed insertion mechanism.

**Spectroscopy Studies**. To probe the mechanism of CS$_2$ adsorption in mm-2–Mg$_2$(dobpdc) in greater detail, in situ diffuse reflectance FT–IR spectra were measured under a pure CS$_2$ atmosphere equal to the vapor pressure at 25 °C (~ 480 mbar; Supplementary Methods). Upon CS$_2$ exposure, two distinct dithiocarbamate C–S resonances are observed at 961 and 949 cm$^{-1}$, which are coupled with the appearance of two resonances at 1343 and 1318 cm$^{-1}$ corresponding to dithiocarbamate C–N single bond vibrations (Fig. 5a). The existence of two discrete resonances for both vibrations suggests the presence of two separate dithiocarbamate structures, in agreement with the two conformations observed via single-crystal X-ray diffraction characterization of CS$_2$-dosed mm-2–Zn$_2$(dobpdc). These assignments are also supported by density functional theory

(DFT) calculations (Supplementary Figures 56–67)[19,31]. Furthermore, the presence of ion-paired ammonium dithiocarbamate chains is supported by a broad resonance at ~2800–1700 cm$^{-1}$, which arises from N–H vibrations of the ion-paired ammonium species (Supplementary Figure 52). In addition to ammonium dithiocarbamate chain formation, significant $CS_2$ condensation is apparent from bands at 2302 and 2164 cm$^{-1}$ (Supplementary Figure 53) [42]. While no chemisorption resonances were observed upon $CS_2$ adsorption in isopentylamine-$Mg_2$(dobpdc) under identical conditions (Fig. 5a), vibrations arising from condensation were found at 2305 and 2165 cm$^{-1}$ (Supplementary Figure 53). Importantly, resonances resulting from condensed $CS_2$ are not observed in the control spectra collected in the absence of the framework (Supplementary Figure 53), indicating that condensation is indeed occurring within the material. For mm-2–$Mg_2$(dobpdc), resonances belonging to condensed $CS_2$ disappeared upon evacuation of the cell, while those arising from the chemisorbed chains remain indefinitely, correlating with hysteretic behavior observed for isothermal adsorption and desorption at 25 °C (Supplementary Figures 54–55). Therefore, as no other discernable $CS_2$-derived species were detected, simultaneous ion-paired chain formation and intra-pore condensation is likely responsible for the higher than expected equilibrium adsorption capacities in the mm-2 appended frameworks (Fig. 3a–c).

To further confirm the structural similarities between the product of reversible $CS_2$ adsorption in mm-2–$Mg_2$(dobpdc) and irreversible adsorption in the Zn analog, in situ near-edge X-ray absorption fine structure (NEXAFS) spectroscopy experiments at the nitrogen K-edge were performed on both materials (Supplementary Methods). All spectroscopic changes were qualitatively reproduced in DFT-computed spectra (Supplementary Methods; Supplementary Table 2; Supplementary Figures 68–71). Upon $CS_2$ dosing, a new pre-edge feature is observed experimentally at 399.8 eV for both frameworks, originating from the nitrogen of a monodentate dithiocarbamate (Fig. 5b). This low energy feature arises from excitation of a core 1 s electron from the trigonal planar nitrogen into the unoccupied $\pi^\star$ dithiocarbamate orbital (Supplementary Figures 72–73; for comparison, $CO_2$ insertion and carbamate formation in N,N′-dimethylethylenediamine–$Mg_2$(dobpdc) resulted in a pre-edge feature arising at 402.3 eV[12], likely due to stronger $\pi$-bonding in the carbamate that leads to a greater degree of $\pi$–$\pi^\star$ splitting and the higher 1 s→$\pi^\star$ transition energy in the NEXAFS spectrum). An ~0.8 eV blueshift of the main-edge feature at 405 eV is also observed for both frameworks, indicative of ammonium formation. Signatures of a new, broad feature in the 410–418 eV range are also evident due to the new C–N bond of the dithiocarbamate. Ex situ NEXAFS data collected at the sulfur K-edge for both frameworks likewise shows strong agreement with predicted spectra for ammonium dithiocarbamate chain formation, with a low energy pre-edge feature again observed and ascribed to coupling to the $\pi^\star$ system of the dithiocarbamate (Supplementary Figures 74–81). Similar changes are observed in the calculated and experimental nitrogen and sulfur K-edge spectra for mm-2–$Mg_2$(dobpdc) and mm-2–$Zn_2$(dobpdc) upon $CS_2$ adsorption, offering further support that ion-paired chain formation upon insertion, as observed in the single-crystal X-ray diffraction structure of mm-2–$Zn_2$(dobpdc), is indeed responsible for the cooperative and reversible adsorption of $CS_2$ in mm-2–$Mg_2$(dobpdc).

## Discussion

Taken together, the foregoing suite of diffraction, computation, and spectroscopic data unambiguously identifies ammonium dithiocarbamate chain formation as the origin of cooperative adsorption of the toxic commodity chemical $CS_2$ in mm-2–$M_2$(dobpdc) (M = Mg, Mn, Zn). Notably, this work represents the extension of a chemisorption process hitherto wholly specific to $CO_2$[12,16]. In contrast to $CO_2$ insertion, $CS_2$ insertion is reversible only for the Mg congener, whereas irreversible adsorption or deleterious thermolysis occurs for the Mn and Zn analogs. Clean, reversible binding of $CS_2$ in mm-2–$Mg_2$(dobpdc) under mild conditions is rather surprising, given the high thermal stability of most reported dithiocarbamate complexes[19,31], and arises from the combination of a weakly thiophilic metal, a rigid square pyramidal coordination geometry enforced by the framework, and ion-pairing with a proximal ammonium cation. The effects of ion-pairing are particularly intriguing, causing reduced charge density on the coordinated sulfur and thus a weakening of the metal–sulfur bond. This result highlights the necessity of both the primary and secondary coordination environments imposed by mm-2–$Mg_2$(dobpdc) in differentiating the resulting dithiocarbamate binding mode from those of irreversibly coordinated molecular dithiocarbamate complexes. Moving forward, this work should serve as a guideline for the exploration of cooperative chemical adsorption of other small molecules in metal–organic frameworks.

## Methods

**General synthesis and characterization methods**. 4,4′-dihydroxy-(1,1′-biphenyl)-3,3′-dicarboxylic acid ($H_4$dobpdc) was purchased from Hangzhou Trylead Chemical Technology Co. and used without further purification. All other reagents and solvents were obtained from commercial suppliers at reagent grade purity or higher and were used without further purification. Ultrahigh purity (99.999%) He and $N_2$ and research grade (99.998%) $CO_2$ were used for all adsorption experiments.

**Surface area measurements**. Langmuir and BET surface area measurements were performing using a Micromeritics ASAP 2420 instrument. In a typical measurement, 50–200 mg of powder was transferred to a pre-weighed glass measurement tube under a $N_2$ atmosphere and capped with a Micromeritics *TranSeal*. The sample was then degassed on the adsorption instrument at the specified activation temperature for 16 hours (outgas rate was less than 3 μbar min$^{-1}$). The evacuated tube was then weighed to determine the mass of the degassed sample. The sample was then fitted with an isothermal jacket and transferred to the analysis port of the adsorption instrument, where the outgas rate was again confirmed to fall below 3 μbar min$^{-1}$. The Langmuir and BET surface areas were measured in a 77 K liquid $N_2$ bath and calculated using the Micromeritics software with a cross-sectional area of 16.2 Å for $N_2$. Argon isotherms were similarly measured using an 87 K liquid argon bath. Pore Size distribution was estimated from 87 K Ar isotherms using non-local density functional theory (NLDFT) implementing a hybrid kernel for Ar adsorption at 87 K based on a zeolite/silica model containing cylindrical pores, as previously described for this family of materials.

**$CO_2$ isothermal adsorption measurements**. Measurements were performing using a Micromeritics ASAP 2420 instrument. In a typical measurement, 50–200 mg of powder was transferred to a pre-weighed glass measurement tube under a $N_2$ atmosphere and capped with a Micromeritics *TranSeal*. The sample was then degassed on the adsorption instrument at the specified activation temperature for 16 h (outgas rate was less than 3 μbar min$^{-1}$). The evacuated tube was then weighed to determine the mass of the degassed sample. The sample was then transferred to the analysis port of the adsorption instrument, where the outgas rate was again confirmed to fall below 3 μbar min$^{-1}$. Isotherms at 25 °C were collected in a water bath. Isotherms at 0 °C were collected in an ice bath.

**Laboratory powder X-ray diffraction**. Laboratory powder X-ray diffraction patterns were collected using a Bruker AXS D8 Advance diffractometer with Cu Kα radiation (λ = 1.5418 Å), a Göbel mirror, and a Lynxeye linear position-sensitive detector, and the following optics: fixed divergence slit (0.6 mm), receiving slit (3 mm), and secondary-beam Soller slits (2.5°). Generator settings were 40 kV and 40 mA. All powder X-ray diffraction patterns were collected at room temperature in air.

**$^1$H NMR spectroscopy of digested metal–organic frameworks**. To analyze amine loading, ~10 mg of amine-functionalized $M_2$(dobpdc) (M = Mg, Zn) powder was digested in a solution of 0.1 mL of 35 wt. % DCl in $D_2$O and 0.6 mL of DMSO-$d_6$. $^1$H NMR spectra were acquired on Bruker AV-300, ABV-400, or AVQ-400

instruments at the University of California, Berkeley NMR facility. Amine loadings were determined from the ratios of the integrated diamine resonances to those of the H$_4$dobpdc ligand.

**Large scale synthesis of M$_2$(dobpdc) (M = Mg, Mn, Zn) Mg$_2$(dobpdc).** An Erlenmeyer flask was charged with Mg(NO$_3$)$_2$·6H$_2$O (5.75 g, 22.5 mmol, 1.24 eq.), 4,4′-dihydroxy-[1,1′-biphenyl]-3,3′-dicarboxylic acid (H$_4$dobpdc; 4.95 g, 18.0 mmol, 1.00 eq.), N,N-dimethylformamide (45 mL), and methanol (55 mL). The mixture was sonicated until all of the solids dissolved. The mixture was filtered through filter paper into a 300 mL screw-cap high-pressure reaction vessel equipped with a stir bar. The reaction mixture was sparged with N$_2$ for 1 h. The reaction vessel was sealed, and the reaction mixture was allowed to stir slowly at 120 °C for 14 h, resulting in precipitation of a white solid from solution. The non-homogenous mixture was filtered, and the solid was quickly transferred to a Pyrex jar filled with N,N-dimethylformamide (250 mL). The jar was placed in an oven heated to 60 °C and allowed to stand for at least 3 h, at which time the jar was cooled to room temperature and the solvent was decanted and replaced with fresh N,N-dimethylformamide (250 mL). The jar was reheated to 60 °C, and this washing process was repeated a total of three times. Next, the N,N-dimethylformamide was replaced with methanol (250 mL), and the washing process was repeated an additional three times using methanol. A small portion of the solid was removed and placed in a vial under flowing N$_2$. The solid was activated under flowing N$_2$ at 250 °C for 24 h, transferred to a glass adsorption tube equipped with a Micromeritics *TransSeal*, and activated for an additional 24 h under high vacuum ( < 10 μbar) at 250 °C. Activated Mg$_2$(dobpdc) was obtained as a white solid. Langmuir surface area (77 K, N$_2$): 4130 m$^2$ g$^{-1}$. BET surface area (77 K, N$_2$): 3250 m$^2$ g$^{-1}$.

**Mn$_2$(dobpdc).** An Erlenmeyer flask was charged with MnCl$_2$·4H$_2$O (990 mg, 5.00 mmol, 2.50 eq.), 4,4′-dihydroxy-[1,1′-biphenyl]-3,3′-dicarboxylic acid (H$_4$dobpdc; 548 mg, 2.00 mmol, 1.00 eq.), N,N-dimethylformamide (100 mL), and ethanol (100 mL). The mixture was sonicated until all of the solids dissolved. The mixture was filtered through filter paper into a 300 mL screw-cap high-pressure reaction vessel equipped with a stir bar. The reaction mixture was sparged with N$_2$ for 1 h. The reaction vessel was sealed, and the reaction mixture was allowed to stir slowly at 120 °C for 14 h, resulting in precipitation of a pale-yellow solid from solution. The non-homogenous mixture was filtered, and the solid was quickly transferred to a Pyrex jar filled with N,N-dimethylformamide (250 mL). The jar was placed in an oven heated to 60 °C and allowed to stand for at least 3 h, at which time the jar was cooled to room temperature and the solvent was decanted and replaced with fresh N,N-dimethylformamide (250 mL). The jar was reheated to 60 °C, and this washing process was repeated a total of three times. Next, the N,N-dimethylformamide was replaced with methanol (250 mL), and the washing process was repeated an additional three times using methanol. A small portion of the solid was removed and placed in a vial under flowing N$_2$. The solid was activated under flowing N$_2$ at 250 °C for 24 h, transferred to a glass adsorption tube equipped with a Micromeritics *TransSeal*, and activated for an additional 24 h under high vacuum ( < 10 μbar) at 250 °C. Activated Mn$_2$(dobpdc) was obtained as a pale yellow solid. Langmuir surface area (77 K, N$_2$): 3500 m$^2$ g$^{-1}$. BET surface area (77 K, N$_2$): 2410 m$^2$ g$^{-1}$.

**Zn$_2$(dobpdc).** A Schlenk flask equipped with a stir bar was charged with ZnBr$_2$·2H$_2$O (8.35 g, 32.0 mmol, 3.20 eq.), 4,4′-dihydroxy-[1,1′-biphenyl]-3,3′-dicarboxylic acid (H$_4$dobpdc; 2.74 g, 10.0 mmol, 1.00 eq.), fresh N,N-dimethyl-formamide (250 mL), and ethanol (250 mL). The mixture was stirred under N$_2$ until all of the solids dissolved. The mixture was sparged with N$_2$ for 1 h. The Schlenk flask was sealed under positive N$_2$ pressure and placed in an oil bath that had been preheated to 120 °C and allowed to stir at this temperature for 14 h, resulting in precipitation of an off-white solid from solution. The reaction mixture was cooled to room temperature and the solid was allowed to settle. The solvent was carefully removed by cannulation under N$_2$ and replaced with fresh, degassed N,N-dimethylformamide (200 mL). The mixture was allowed to stand for at least 24 h, at which time the Schlenk flask was cooled to room temperature, the solid was allowed to settle, and the solvent was removed by cannulation and replaced with fresh, degassed N,N-dimethylformamide (200 mL). The flask was reheated to 60 °C, and this washing process was repeated a total of three times. Next, the N,N-dimethylformamide was replaced with degassed methanol (200 mL), and the washing process was repeated an additional three times using methanol. The solvent was removed under high vacuum at 250 °C in the Schlenk flask. The flask was transferred to a N$_2$-filled glovebox, and the solid was transferred to a glass adsorption tube equipped with a Micromeritics *TransSeal*. The sample was activated for an additional 24 h under high vacuum ( < 10 μbar) at 250 °C to yield Zn$_2$(dobpdc) as an off-white solid. Langmuir surface area (77 K, N$_2$): 2770 m$^2$ g$^{-1}$. BET surface area (77 K, N$_2$): 2150 m$^2$ g$^{-1}$.

**Synthesis of mm-2–Mg$_2$(dobpdc).** A 20 mL scintillation vial was charged with 10 mL of toluene and 3 mL of mm-2 (N,N-dimethylethylenediamine, purchased from Sigma–Aldrich). Methanol-solvated Mg$_2$(dobpdc) (∼100 mg) was filtered and washed with successive aliquots of toluene (2 × 10 mL). Note: Mg$_2$(dobpdc) should not be allowed to dry completely as this can in some cases lead to decomposition of the framework. Next, Mg$_2$(dobpdc) was added to the mm-2/toluene solution, and

the vial was swirled several times and allowed to stand at room temperature for 24 h. The mixture was then filtered, and the resulting powder was thoroughly washed with toluene (3 × 20 mL) and allowed to dry on the filter paper for 2 min. The isolated powder was activated at 100 °C for 2 h under flowing N$_2$ to remove excess mm-2 and toluene from the pores. $^1$H NMR spectroscopy of the digested material confirmed the loading of mm-2 to be ∼100% (Supplementary Figure 6).

**Synthesis of isopentylamine–Mg$_2$(dobpdc).** A 20 mL scintillation vial was charged with 10 mL of toluene and 3 mL of isopentylamine (purchased from Sigma–Aldrich). Methanol-solvated Mg$_2$(dobpdc) (∼100 mg) was filtered and washed with successive aliquots of toluene (2 × 10 mL). Note: Mg$_2$(dobpdc) should not be allowed to dry completely as this can in some cases lead to decomposition of the framework. Next, Mg$_2$(dobpdc) was added to the isopentylamine/toluene solution, and the vial was swirled several times and allowed to stand at room temperature for 24 h. The mixture was then filtered, and the resulting powder was thoroughly washed with toluene (3 × 20 mL) and allowed to dry on the filter paper for 2 min. The isolated powder was activated at 100 °C for 2 h under flowing N$_2$ to remove excess isopentylamine and toluene from the pores. $^1$H NMR spectroscopy of the digested material confirmed the loading of isopentylamine to be ∼100% (Supplementary Figure 10).

**Synthesis of mm-2-M$_2$(dobpdc) (M = Mn, Zn).** Methanol-solvated M$_2$(dobpdc) (∼ 100 mg) was filtered and washed with successive aliquots of toluene (2 × 10 mL). Note: M$_2$(dobpdc) (M = Mn, Zn) should not be allowed to dry completely as this can in some cases lead to decomposition of the framework. The isolated powder was transferred to a 40 mL scintillation vial, and the solvent was removed under flowing N$_2$ at 250 °C for 16 hours. A solution of 10 mL of toluene and 3 mL of mm-2 was dried by stirring over CaH$_2$ for 3 h at 60 °C. After centrifugation of the CaH$_2$ suspension and cooling of the activated framework to room temperature under N$_2$, the mm-2/toluene solution was transferred via syringe onto the activated powder under positive pressure of N$_2$. The suspension was transferred into a N$_2$ glove tent and left undisturbed overnight. In the glove tent, the mixture was then filtered, and the resulting powder was thoroughly washed with dry toluene (3 × 20 mL) and allowed to dry on the filter paper for 2 min. The isolated powder was activated at 100 °C for 2 h under flowing N$_2$ to remove excess mm-2 and toluene from the pores. $^1$H NMR spectroscopy of the digested Zn congener confirmed the loading of mm-2 to be ∼100% (Supplementary Figure 15).

**CS$_2$ isothermal adsorption measurements.** In a typical isotherm measurement, 100–200 mg of powder was transferred to a pre-weighed glass measurement tube and capped with a Micromeritics *TransSeal*. Importantly, all O-rings used in the presence of CS$_2$, including on the *TransSeals* and analysis ports, were composed of Kalrez, a CS$_2$ resistant perfluoroelastomer. The sample was then degassed on an activation port of a Micromeritics 2420 adsorption instrument at the specified activation temperature for 16 hours (outgas rate was less than 3 μbar min$^{-1}$). The evacuated sample tube was then weighed to determine the mass of the degassed sample. The sample was transferred to the analysis port of the Micromeritics 3Flex Surface Characterization instrument, where the outgas rate was again confirmed to fall below 3 μbar min$^{-1}$. Note: mm-2–Mg$_2$(dobpdc) samples could be re-activated for reuse, whereas the Mn and Zn congeners could not. A volume of 10 mL of anhydrous CS$_2$ ( ≥ 99%) purchased from Sigma-Aldrich was transferred to a 20 mL oven-dried steel bomb and loaded onto the 3Flex instrument. The CS$_2$ sample was subjected to three freeze–pump–thaw cycles on the instrument before use. The CS$_2$ sample holder was maintained at room temperature throughout all isothermal measurements, thus remaining a volatile liquid. Isothermal measurements from 25–80 °C were conducted using a water circulator to control the temperature of a Syltherm XLT silicone oil bath. From 90–120 °C, isothermal measurements were conducted using a silicone oil bath, with temperature controlled by a hot plate possessing a thermocouple probe. The temperature of the bath was cross-referenced with a secondary external digital temperature probe. In order to achieve reproducible results, an equilibration time of 240 s was required and thus used for all measurements. Adsorption equilibrium was assumed when the variation of pressure was 0.01% or lower over 240 seconds. Differential enthalpies of adsorption were determined using the Clausius–Clapeyron equation at constant loading using linear interpolation with a 1$^{st}$ order spline for isotherms at three different temperatures. Plotting the corresponding pressures and temperatures as ln $P$ vs 1/$T$ (K$^{-1}$) afforded the differential entropies of adsorption from the y-intercept of the resulting lines. The Clausius–Clapeyron relationship is expressed as:

$$\ln P = \frac{\Delta h_{ads}}{RT} + C$$

Where $P$ is the pressure, $T$ is the temperature, $R$ is the universal gas constant, and $C$ is a constant equal to $-\Delta s_{ads} \, R^{-1}$.

## Data Availability

The authors declare that all data supporting the findings of this study are available within the paper (and its supplementary information files). The X-ray crystallographic coordinates for structures reported in this study have been

deposited at the Cambridge Crystallographic Data Centre (CCDC), under deposition number 1834399. These data can be obtained free of charge from The Cambridge Crystallographic Data Centre via www.ccdc.cam.ac.uk/data_request/cif.

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

## Acknowledgements

This work was supported by the Center for Gas Separations Relevant to Clean Energy Technologies, an Energy Frontier Research Center funded by the U.S. Department of Energy, Office of Science, Office of Basic Energy Sciences under Award DE-SC0001015. We thank the Philomathia Foundation and Berkeley Energy and Climate Institute for support of C.M.M through a postdoctoral fellowship and the National Institutes of Health for support of P.J.M. through a postdoctoral fellowship (GM120799). The use of Advanced Light Source beamline 11.3.1 for the collection of single-crystal X-ray diffraction data, the use of Advanced Light Source beamline 6.3.2 for in situ N K-edge XAS, the use of Advanced Light Source beamline 10.3.2 for ex situ S K-edge XAS (with guidance from Dr. Sirine Fakra), a user project at The Molecular Foundry (facilitated by L.F.W. and D.P.) and use of its computer clusters vulcan and etna managed by the High Performance Computing Services Group, and the use of the National Energy Research Scientific Computing Center were performed at Lawrence Berkeley National Laboratory, which is supported by the Director, Office of Science, Office of Basic Energy Sciences, of the DOE under contract no. DE-AC02-05CH11231. Powder X-ray diffraction data were collected at the Advanced Photon Source, a U.S. Department of Energy (DOE) Office of Science User Facility operated for the DOE Office of Science by Argonne National Laboratory under Contract No. DE-AC02-06CH11357. DFT infrared vibrations calculations were performed at the MGCF in the College of Chemistry at the University of California, Berkeley, a facility funded by NIH S10OD023532, with guidance provided by Dr. Olayinka Olatunji-Ojo. We thank Dr. Katie R. Meihaus for editorial assistance.

## Author Contributions

C.M.M. and J.R.L. formulated the project. C.M.M. and P.J.M. synthesized the compounds. C.M.M. characterized the compounds. C.M.M., D.A.R., and M.I.G. collected and analyzed the gas adsorption data. R.L.S. collected and analyzed the single-crystal X-ray diffraction data. C.M.M., W.S.D., and G.M.S. collected X-ray absorption data. W.S.D. and G.M.S. analyzed the X-ray absorption data, with associated calculations by L.F.W. and D. P. J.O. collected the powder X-ray diffraction data. T.R. analyzed the powder X-ray diffraction data. H.Z.H.J. collected the infrared data. C.M.M. performed the vibrational

calculations. C.M.M. and J.R.L. wrote the paper, and all authors contributed to revising the manuscript.

## Additional information

**Competing interests:** The authors declare no competing interests.

