## [Peer Review File · Nature Communications]

Reviewers' comments:

Reviewer #1 (Remarks to the Author):

This manuscript describes a comprehensive study of adsorption, binding, and reversibility tests of CS₂ in three amine-appended isostructural MOFs, namely Mg, Zn and Mn-MOFs. Adsorption isotherms of CS₂ have been tested in all three MOFs and Mg-MOF has been found to show excellent reversibility whereas the other two undergo irreversible chemical sorption. The host-guest binding mechanism and location of adsorbed CS₂ molecules have been determined from X-ray experiments, IR, and X-ray absorption spectroscopy. DFT modelling provided theoretical insight where necessary. This work builds upon the recent work on CO₂ binding from the same group (ref8) and is largely incremental in nature. That said, I do find the work was carried out in sufficient manner and given the rarely explored CS₂ in MOFs in general, I recommend for publication following the revisions below.

1. Isotherms: the boiling point of CS₂ is around 46 C and the reported adsorption isotherms should be provided in P/P₀ form to allow a better, direct comparison throughout. Figure 3b shows that the MOF does not adsorb water (or very little) at 75 C. is this true? I had believe there was strong adsorption of water in this class of MOFs. I would like to see a standing alone water adsorption for these MOFs.

2. Following to above, since CS₂ is a liquid at room temperature, I am not entirely certain what "capture application" is actually targeted for the present study? this needs to be clarified using, ideally, practical conditions and/or real world application examples. If CS₂ is to be captured at low concentration (I assume it is), some breakthrough experiment at relevant concentration is required to support the central claim, at least for the Mg-MOF which shows reversibility.

3. The distinct reversibility of the Mg-MOF needs further investigation. Figure 3c suggests Mg and Zn-MOFs are rather similar, whereas Mn-MOF shows stronger adsorption. This is different to that observed for the reversibility.

4. X-ray experiment: Figure 4 shows the product is a mixture of two phases with 50% and 40% each. what is the other 10%? is that the bare MOF phase? Similarly in SI, it is also a mixture of some 40 and 30% except for one system where only one conformation was observed. The actual sample is composed of up to 3 phases; I am rather surprised that all three phases have identical lattice parameters. Full refinements of structures with this level of complexity is over-kill for PXRD data, particularly for Mg-MOF data where only a limited number of peaks were observed. the CIFs derived from XPRD refinements need to be reported. Overall, more details on the structural refinement and treatment of the model need to be provided.

Reviewer #2 (Remarks to the Author):

Review Report for Nature Communications (NCOMMS-18-12050-T)

The manuscript reports the cooperative adsorption of carbon disulfide in diamine-appended Metal–Organic Frameworks (MOFs). The significant adsorption of carbon disulfide in the N,N-dimethylethylenediamine appended M₂(dobpdc) (M = Mg, Mn, Zn) have been verified using adsorption isotherms, FTIR analyses, X-ray diffraction studies and spectroscopic characterizations. The previously reported diamine-appended M₂(dobpdc) (M = Mg, Mn, Zn) MOFs have been studied for CS₂ adsorption systematically (like CO₂ adsorption) and all the hypothesis have been supported experimentally. This reviewer thus believe that a suitably revised version of this study could be interesting for the readers of the Nature Communications, however only after the

following aspects have been fully addressed:

1. Authors have mentioned that in determining viable candidate adsorbates, three necessary properties such as having a dipole or quadrupole moment, formation of coordinatively stable Brønsted acidic species and formation of the bonds with the amine and metal are mandatory. However, it is not clear that what is role of coordinatively stable Brønsted acidic species in this adsorption process. Authors should elaborate this in the revised manuscript.
2. It has been showed in manuscript that diamine plays a central role in promoting the observed adsorption behavior and N,N-dimethylethylenediamine appended M₂(dobpdc) (M = Mg, Mn, Zn) shows superior performance as compared to isopentylamine appended MOFs. Is the free amine groups responsible for the higher adsorption capability in the N,N-dimethylethylenediamine appended MOFs?
3. How would be the performance of N,N-Diethylethylenediamine appended M₂(dobpdc) (M = Mg, Mn, Zn) MOFs for CS₂ adsorption? Increasing the appending group size, apparently decreasing the pore size, will have any effect on the adsorption capability?
4. In the manuscript, it has been hypothesized that intrapore CS₂ condensation occurs along with chemisorption in MOFs resulting in the irreversible sorption process, which can be reversible with mild thermal input. However, in an ideal adsorption process, it is expected to have a completely physisorptive adsorption process for facile usages at industrial scale, without any kinetic or thermal inputs. In these circumstances, adsorption process does not seem to completely reversible at atmospheric conditions.
5. Since M₂(dobpdc) (M = Mg, Mn, Zn) have open metal sites, is there any adsorption of CS₂ in amine-free frameworks?
6. As confirmed from the adsorption isotherms, only Mg(dobpdc) framework is capable of reversible CS₂ adsorption. Author's need to provide more insights using simulations in all the M₂(dobpdc) (M = Mg, Mn, Zn) to explore the role of electronic factors involved in this adsorption process, where Mg(dobpdc) shows reversible adsorption and Zn/Mn (dobpdc) show reversible CS₂ adsorption.

As mentioned in the aforementioned comments, this reviewer feels this manuscript needs a major revision, before considering it for publication in Nature Communications.

Reviewer #3 (Remarks to the Author):

The manuscript from Long et al reports the adsorption of carbon disulphide in metal-organic frameworks (MOF) isoreticular to the well-known MOF-74/CPO-27, modified with diamines in the accessible, unsaturated open metal sites. This work is based on their previous study published in Nature, about CO₂ adsorption, and similar papers published by the authors on this family of materials. The whole paper tries to raise the excitement of the materials with statements about the uniqueness of the cooperative adsorption mechanism of these MOFs. However, although there is a lot of work and hours included in the manuscript, I cannot share the excitement in terms of adsorption performance and novelty, as this kind of behavior/performance in porous materials has been shown many times before. The quality in terms of adsorption characterization is also not adequate. The authors insist in the fact that "importantly, cooperative adsorption profiles allow for greater working capacities with smaller temperature swings", and although I, of course, agree with this idea, I ended up with some different conclusions, as described by the end of this report. All in all the study is very complete, but I have not seen any new techniques (either a combination of) that have not been used before in adsorption studies of many other gases and molecules. This includes crystallography, spectroscopy, gas adsorption analysis, DFT calculations as well as other ones that could be included (calorimetry, Monte Carlo, etc.). Indeed, the technique I know more, gas adsorption, is far away from well-established standards – this includes the fittings, range of pressure for N₂ adsorption, Langmuir surface area, pore size distribution (missing), etc. I hope the other techniques and DFT calculations are better. For all these, and the reasons detailed below, I do not recommend this paper to be published in Nature Communications.

- Figure 1 is a typical textbook representation of adsorption isotherms and the importance of working capacity. I am not sure if this should be included, but now that is included, the representation of the working capacity in a temperature swing process is wrong. If the process is isobaric, $P_{ads} = P_{des}$. Of course, it can be argued that if the adsorber chamber is isolated and closed, heating up will imply an increase of the pressure... but this is not the idea, as it will not allow adsorbent regeneration. As the authors do correctly later on, they obtained the working capacity in an isobaric process, where the working capacity is higher – I have some questions though, as the relative pressure could have gone to zero.

- Page 3. “metal-organic frameworks that bind small molecules in a cooperative fashion show minimal guest uptake below a temperature-dependent threshold pressure, followed by a sudden and sharp rise, or step, in adsorption (a Type V adsorption isotherm; Fig. 1b)^{8,9}”. Reference 9 is a completely different behavior to what the authors include in this sentence, as this is not cooperative adsorption as such, but rather a breathing mechanism found in soft porous materials such as MIL-53. In any case, I agree the shape of the isotherm is similar – but this is my point, this adsorption performance (shape of the isotherms) is not new and has been described many times in the past 10-20 years.

- Page 4. “we recently discovered the first example of a material displaying cooperative adsorption via a unique, chain-forming mechanism⁸”. The paper was published in Nature – fair enough. However, this kind of behavior/performance is not as novel as the authors are suggesting, and is indeed similar to many gate opening mechanisms reported in the last 10-20 years in the field of porous coordination polymers (breathing, gate-opening). Of course, the adsorption mechanism is not the same, but the performance due to the reorganization of amines (or other organic entities in PCPs) is not unique nor novel. Hill coefficients have indeed been used for adsorption isotherms long time ago – even outside of the MOF community (see for example the work from Kanoh and Kaneko). The shape of the isotherms is common enough to be included in the IUPAC classification since the last century – I would not define this unique and unprecedented.

- Page 4: “Despite the obvious practical advantages, thus far all investigations of this cooperative process have been limited to CO₂ adsorption^{8,10,11}” I completely disagree with this statement. A typical example about MOFs on this are the recent works on water adsorption from e.g. Yaghi et al. What is true is that cooperative interaction typically takes place with molecules with high intermolecular interaction – so it is more difficult to happen with e.g. methane at room temperature, although it has been described for this gas molecule at much lower temperatures (eg 125K, Yildirim et al.) or by the same authors of the current manuscript three years ago using, again, a flexible structure.

- Page 5. “Microcrystalline powders of M₂(dobpdc) (M = Mg, Mn, Zn) were prepared and characterized according to previously published procedures”. Measuring Langmuir surface area for a microporous material in this century is simply wrong. This was done broadly by the MOF community when these materials were discovered, but the results are not adequate since they are only valid to give higher values for the specific surface area. This is well described in the recommendations from the IUPAC, also for MOFs.

The N₂ adsorption isotherms reported in the supplementary information are extremely far away from the quality expected for an academic journal. In MOF-74-like MOFs, the adsorption takes place at very low relative pressures, and this is where the data needs to be compared (as is done for all the other gases reported). The reported data gives information about the pore volume, but there is no comparison on the very low-pressure range (0-0.01 p/p₀), or the pore size distribution. More importantly, if N₂ saturation takes place at ca 760 mbar, how do the authors reach 1000

mbar? The authors have a 3Flex available that they should use always.

- Page 6 "Using linear spline fits and the Clausius–Clapeyron relationship". Linear spline is a technique for interpolation rather than fitting. In this case, looking at eg Sup. Figure 23, there is some obvious noise in the jump due to equilibration issues, but there is no fitting here but an interpolation. Of course, a proper fitting needs to be done (there are multiple equations in the literature), and the Clausius-Clapeyron representation should be included in the ESI.

"When CS₂ adsorption was measured for Mg₂(dobpdc) appended with isopentylamine, a primary monoamine with a similar steric profile to mm-2, only Langmuir-type adsorption was observed with a Δh_{ads} of just -26.0 ± 0.3 kJ/mol (Supplementary Fig. 26)." Again, these isotherms are not Type I since they show some steps – this can be demonstrated by trying the fitting of a Langmuir equation instead of the spline interpolation included in Fig. S27.

"Impressively, individual isotherms of various gases at 75 °C, including CO₂, suggest a near-perfect selectivity for CS₂ (Fig. 3b)." This is true and expected for molecules smaller and with lower guest-guest interactions than CS₂, but not for water. Why was the isotherm measured up to 30 mbar only and not 1000 mbar as the others?

- Figure 3. Why is the isotherm for mm-2–Mg₂(dobpdc) at 25C different in Fig. 3a and Fig. 3c? At 100 mbar one reaches 8 mmol/g but the other only 4 mmol/g. Also, for Fig. 3c there is a point of the isotherm (6 mbar) significantly displaced.

Also, the desorption isotherms are only provided for 25C – this should be done for the whole range of temperatures or, at least, for 100C (Mg-MOF), to see how reversible the equilibrium is. This would help to clarify the methods employed in the analysis during the temperature swing adsorption process. For how long is the reactivation done? Why is the CO₂ included during the cooling down instead of using the N₂ flow? Also, if I understood correctly, these are not isobars, because there is no CS₂ during the heating, so the relative pressure is zero – is this correct?

- Finally, I found the 6.8 mmol/g working capacity of mm-2-Mg₂(dobpdc) interesting. The process takes advantage of the s-shape of the isotherms. However, the presence of the mm in the porosity also limits the pore volume and adsorption capacity. In this regard, the authors do not mention an easier alternative they have in their results: non-modified Mg₂(dobpdc) shows a capacity of ca. 19 mmol/g at 25C and 75mbar, and an loading of ca 6 mmol/g at 75C and 75 mbar (I suspect much lower at 100C, say 4mmol/g) – the isotherms being fully reversible at these temperatures. This translates to working capacities of 13-15 mmol/g, twice or three times higher than the ones reported in the modified MOFs and without any issues on reversibility. I am wondering if I missed something important here.

- Page 16: "Moving forward, this work should serve as a guideline for the exploration of cooperative adsorption of other small molecules in metal-organic frameworks." Again, I do not share this excitement, as there are many important issues that would not define this work as a guideline.

Response to Reviewers:

Manuscript ID: NCOMMs-18-12050-T

Title: "Cooperative Adsorption of Carbon Disulfide in Diamine-Appended Metal–Organic Frameworks"

Authors: C. Michael McGuirk, Rebecca L. Siegelman, Walter S. Drisdell, Tomče Runčevski, Phillip J. Milner, Julia Oktawiec, Liwen F. Wan, Gregory M. Su, Henry Z. H. Jiang, Douglas A. Reed, Miguel I. Gonzalez, David Prendergast, Jeffrey R. Long

Reviewer #1:

We thank the reviewer for the careful reading and helpful comments.

1. Isotherms: the boiling point of CS₂ is around 46 C and the reported adsorption isotherms should be provided in P/P₀ form to allow a better, direct comparison throughout. Figure 3b shows that the MOF does not adsorb water (or very little) at 75 C. is this true? I had believe there was strong adsorption of water in this class of MOFs. I would like to see a standing alone water adsorption for these MOFs.

Thank you for the suggestion. We have re-plotted Figure 3a as P/P_0 , included this new plot in the supporting information (Figure S28), and referenced it in the main body of the manuscript. For the main text, we believe that an absolute pressure scale (as it currently stands) is more approachable for the general audience of *Nature Communications*.

Yes, the reported water isotherm at 75 °C is as measured. For water isotherms, the maximum pressure is limited by the saturation pressure at the coldest point in the instrument. As a result, we are restricted to a maximum pressure of about 30 mbar ($P/P_0 = 0.08$) at 75 °C to avoid condensation in areas of the instrument that cannot be heated. Of note, we have previously reported a full set of water isotherms at various temperatures for mm-2–Mg₂(dobpdc)¹. These data do show water condensation at higher relative pressures, as observed in the lower temperature water isotherms. However, diamine-appended frameworks with exposed tertiary amines, such as the mm-2-appended framework discussed here, were generally found to show diminished water adsorption compared to those with exposed primary or secondary amines. Importantly, water is not able to participate in the cooperative chemical adsorption process, due to the chemical specificity of the reactive adsorption mechanism. The importance of this chemical specificity is emphasized in the updated manuscript. We also note that water adsorption in the diamine-appended frameworks is significantly weaker than that in the "bare" or non-diamine-appended version of these frameworks, where the coordinatively unsaturated metal centers serve as strong binding sites for water.

2. Following to above, since CS₂ is a liquid at room temperature, I am not entirely certain what "capture application" is actually targeted for the present study? this needs to be clarified using, ideally, practical conditions and/or realworld application examples. If CS₂ is to be captured at low concentration (I assume it is), some breakthrough experiment at relevant concentration is required to support the central claim, at least for the MgMOF which shows reversibility.

CS₂ is a highly volatile, toxic liquid with a vapor pressure of ~480 mbar at 25 °C, that is used broadly across many industries. While this indeed motivates a desire to explore adsorbents capable of selectively adsorbing this toxic compound, it is important to note, as emphasized in the manuscript, that

this study is a fundamental investigation of the extension a promising cooperative chemical adsorption mechanism. We include a cycling experiment to demonstrate the ability to achieve stable cycling capacities with the diamine-appended Mg framework; however, we do not intend this paper to suggest immediate deployment of the adsorbent in a specific separation. A more in-depth investigation involving optimization of the cycle configuration as well as breakthrough experiments and selectivity calculations is beyond the scope of this mechanistic investigation and would be more suitable for a follow-up study. In addition, further adsorbent optimization by, for example, changing the appended diamine may enable targeted capture of CS₂ at more industrially relevant conditions. We note that mm-2-Mg₂(dobpdc) cooperatively captures CS₂ at absolute pressures almost two orders of magnitude below the vapor pressure of this volatile liquid at room temperature. Therefore, this material has the potential to remove low partial pressures of CS₂ from the atmosphere during processes that employ liquid CS₂, such as the xanthation of cellulose in the production of rayon and cellophane, the predominate use of CS₂, which is referenced in the introduction.

3. The distinct reversibility of the MgMOF needs further investigation. Figure 3c suggests Mg and ZnMOFs are rather similar, whereas MnMOF shows stronger adsorption. This is different to that observed for the reversibility.

We believe that the observed trends in threshold pressure and reversibility in this chemisorption process can be logically elucidated from well-understood trends in coordination chemistry and thiophilicity, as referenced throughout the manuscript. In light of the Reviewer's comment, we have added further clarification in the main text in order to increase transparency on these points. A full explanation can be found below.

The observed threshold pressures for insertion and subsequent cooperative adsorption are the result of a competition between the stability of the initial octahedral metal centers and the thiophilicity of the given metal²⁻⁴. Previously, we have shown that for CO₂ adsorption in analogous frameworks, the Mg analog has a much lower threshold pressure than Mn and especially Zn². This is because Mg has the lowest octahedral complex stability and the highest oxophilicity, and thus the greatest driving force for insertion into the M–N bond and formation of a M–O bond. Conversely, Zn has the most stable octahedral complex and the lowest oxophilicity in this family, thus the lowest driving force for insertion into the M–N bond, resulting in a much higher threshold pressure. Mn falls between the two. Herein, the distinct deviation from the trend observed for CO₂ is the result of the inverted thiophilicity trend, where Zn is the most thiophilic and Mg is the least⁴. Therefore, the trends in the stability of the initial M–N bond and the desire to form a M–S bond work against each other, resulting in all three metals showing quite similar threshold pressures. Similarly, the distinct reversibility observed for mm-2-Mg₂(dobpdc) is the result of the appreciably lower thiophilicity of Mg in comparison with Mn and Zn⁴. As with adsorption, desorption is a chemical process. During desorption the newly formed dithiocarbamate ligand must be removed from the metal center through the breaking of the M–S bond. Indeed, the low reversibility of dithiocarbamate ligands on first-row transition metals resulting from their high thiophilicity has been broadly demonstrated, so the irreversibility of adsorption on mm-2-Mn₂(dobpdc) and mm-2-Zn₂(dobpdc) is not surprising⁵. In contrast, the Mg–S bond formed upon CS₂ adsorption in mm-2 Mg₂(dobpdc) can be broken with mild thermal input owing to the poor thiophilicity of Mg.

4. Xray experiment: Figure 4 shows the product is a mixture of two phases with 50% and 40% each. what is the other 10%? is that the bare MOF phase? Similarly in SI, it is also a mixture of some 40 and 30% except for one system where only one conformation was observed. The actual sample is composed of up to 3 phases; I am rather surprised that all three phases have identical lattice parameters. Full refinements of structures with this level of complexity is overkill for PXRD data, particularly for MgMOF data where only a limited number of peaks were observed. the CIFs derived from XPRD refinements need to be reported. Overall, more details on the structural refinement and treatment of the model need to be provided.

For the single-crystal X-ray diffraction structure of CS₂-inserted mm-2-Zn₂(dobpdc), the 10% of zinc(II) sites in the framework without dithiocarbamate contain a mixture of solvent, water, CS₂, or unreacted diamine that could not be modeled. We have added this clarification to the figure caption of Figure 4. This explanation is also included in the refinement details in the original supporting information and the .cif itself:

“Refinement of each of the disordered residues resulted in chemical occupancies of 47.5(7)% (part -1, atoms with suffix A in the cif) and 39.9(7)% (part -2, atoms with suffix B in the cif), which were fixed in the final refinement. The overall ammonium dithiocarbamate occupancy is 87.4%; however, the sulfur atom bound to Zn (S1) was kept at fully occupancy to account for solvent, water, CS₂, or unreacted diamine that is bound on sites where the chains are absent. The reported formula, however, reflects the sulfur content for the ammonium dithiocarbamate chains alone.”

The X-ray diffraction experiments give only an average picture of all unit cells over the course of the experiment. Because the symmetry of the structure remains unchanged following CS₂ adsorption, the asymmetric unit contains only half of the ligand, a single metal site, and two partial-occupancy conformations of an ammonium dithiocarbamate unit that branch from the same metal-bound sulfur atom. This high symmetry results from a lack of communication across multiple channels, precluding long-range order that might enable direct analysis of the relative orientations of the two dithiocarbamate conformations. Instead, their relative orientation must be inferred by examining the manner in which the chains pack within the pore. For example, atoms in adjacent chains of the secondary conformation (39.9(7)%) appear to overlap in the *a/b* plane. This physical impossibility implies that adjacent chains must exhibit different or alternating conformations. (Adjacent ammonium dithiocarbamate units of different conformations do not clash with one another in the *a/b* plane.) Furthermore, very close apparent NH...S contacts (1.587(5) Å) for ammonium dithiocarbamate chains constructed from the secondary conformation suggest that alternating conformations of dithiocarbamate units may be possible down the *c* axis as well. As a result, the lattice parameters reflect this average packing of both dithiocarbamate conformations within each pore. We have added a note of this ambiguity in the distribution of the individual conformations in the manuscript.

As the reviewer pointed out, full refinement of structures with this level of complexity using PXRD data is challenging and provides a limited level of structural information. In our analyses, we kept the atomic fractional coordinates fixed, and we freely refined the lattice parameters, profile parameters (strain, stress, instrumental parameters, simple axial model parameter and zero-error shift), background parameters (Chebyshev polynomial of 10th order) and site occupancies of the dithiocarbamate moieties,

as described in detail in the SI. With this approach, we could not extract precise information about bond lengths, angles, and similar structural details; however, the limited refinement serves as strong evidence for structural similarity between the different metal analogues. The differences in site occupancies of different dithiocarbamate moieties can be ascribed to experimental factors (different amount of appended amines upon loading, different level of structural disorder, etc.). We did not report CIFs of these structures because the fractional atomic coordinates are already reported for the Zn analogue, and the refined unit cell parameters (and figures-of-merit) for each metal analogue are given in the SI.

Reviewer #2:

We thank the reviewer for the careful reading and helpful comments.

1. Authors have mentioned that in determining viable candidate adsorbates, three necessary properties such as having a dipole or quadrupole moment, formation of coordinatively stable Brønsted acidic species and formation of the bonds with the amine and metal are mandatory. However, it is not clear that what is role of coordinatively stable Brønsted acidic species in this adsorption process. Authors should elaborate this in the revised manuscript.

We have added a clarification to the introduction on this point. Thank you for the suggestion.

As stated in the introduction, the cooperativity of the previously reported mechanism for CO₂ adsorption in diamine-appended M₂(dobpdc) results from the insertion of CO₂ into the metal–amine bonds coupled with proton transfer to the proximal (non-metal bound) amine, generating one-dimensional, electrostatically paired ammonium-carbamate chains. This insertion into the metal–amine bond and formation of the new coordinating species (i.e., carbamate for CO₂) results in a chemically specific mechanism. This proton transfer is necessary for site-to-site communication, and thus cooperativity will only occur if the chemical species resulting from the insertion into the metal–amine is Brønsted acidic. Thus, for a new molecular species (i.e., CS₂) to show similar cooperativity upon insertion into the metal–amine bond, Brønsted acidity is critical.

2. It has been showed in manuscript that diamine plays a central role in promoting the observed adsorption behavior and N,Ndimethylethylenediamine appended M₂(dobpdc) (M = Mg, Mn, Zn) shows superior performance as compared to isopentylamine appended MOFs. Is the free amine groups responsible for the higher adsorption capability in the N,Ndimethylethylenediamine appended MOFs?

Yes, the presence of two amine groups at each metal center (one coordinated and one free) is central to the observed cooperative mechanism, resulting in step-shaped isotherms. As can be seen in the supplementary information, isopentylamine-appended Mg₂(dobpdc) does *not* show step-shaped adsorption, but rather only weak physisorption (as confirmed by IR experiments). The necessity of the second (free) amine group in N,N-dimethylethylenediamine goes back to the above discussion. For cooperativity—and thus step-shaped adsorption—to occur, site-to-site communication is required. This requirement is satisfied through proton transfer from the acidic coordinating species to the basic free amine. Without that amine, the cooperative mechanism is inaccessible, as is evident from the adsorption and spectroscopic measurements of isopentylamine-appended Mg₂(dobpdc).

3. How would be the performance of N,NDiethylethylenediamine appended M₂(dobpdc) (M = Mg, Mn, Zn) MOFs for CS₂ adsorption? Increasing the appending group size, apparently decreasing the pore size, will have any effect on the adsorption capability?

We thank the reviewer for this perceptive question. Within the context of CO₂ adsorption in these materials, our group has undertaken a considerable effort to explore and characterize the effects of changing the substitution pattern and groups of the diamine¹. The previous work with CO₂ demonstrates that larger alkyl groups on the free amine disrupt the close pairing of the resulting cation (ammonium) and anion (carbamate) upon insertion, thus lowering the strength of this interaction and comparatively

disfavoring insertion. The functional result of this is that the *N,N*-diethylethylenediamine-appended framework has a higher threshold pressure for cooperative adsorption at a given temperature. We would expect a very similar trend for CS₂, where *N,N*-diethylethylenediamine would yield adsorption steps at higher threshold pressures. Additionally, we may expect a slightly lower gravimetric capacity, as the larger alkyl groups are heavier and leave less accessible pore volume for CS₂ condensation upon ammonium dithiocarbamate chain formation. A similar broad exploration of the effects of diamine substitution on cooperative CS₂ adsorption would be fitting for a standalone follow-up study.

4. In the manuscript, it has been hypothesized that intrapore CS₂ condensation occurs along with chemisorption in MOFs resulting in the irreversible sorption process, which can be reversible with mild thermal input. However, in an ideal adsorption process, it is expected to have a completely physisorptive adsorption process for facile usages at industrial scale, without any kinetic or thermal inputs. In these circumstances, adsorption process does not seem to completely reversible at atmospheric conditions.

We agree that the adsorption and desorption kinetics are critical to consider for potential application of these materials, and this is an ongoing area of investigation in our group. However, we respectfully disagree with the reviewer regarding the desirable thermodynamics of adsorption. While purely physisorptive materials may enable facile regeneration, materials with strong and adsorbate-specific binding sites are necessary to achieve high chemical selectivity at low partial pressures. See for example *Energy Environ. Sci.*, **2011**, *4*, 3030 for a demonstration of the advantage of strong and selective binding sites in CO₂ capture⁶. A temperature or pressure swing will ultimately be required for a necessarily selective sorbent⁷, and our aim in exploring unique mechanisms such as that detailed here is the realization of adsorption processes that minimize the energy consumption of that swing^{1,2,8}. Through a cooperative and chemically selective adsorption mechanism, the diamine-appended framework mm-2-Mg₂(dobpdc) affords high selectivity for CS₂ while minimizing the temperature swing needed to achieve the full working capacity of the material.

5. Since M₂(dobpdc) (M = Mg, Mn, Zn) have open metal sites, is there any adsorption of CS₂ in aminefree frameworks?

Diamine-free Mg₂(dobpdc) shows Type IV adsorption, with physical adsorption at the open metal sites followed by capillary condensation. This CS₂ adsorption data for Mg₂(dobpdc) can be found in the supporting information (Supplementary Figure 33).

6. As confirmed from the adsorption isotherms, only Mg(dobpdc) framework is capable of reversible CS₂ adsorption. Author's need to provide more insights using simulations in all the M₂(dobpdc) (M = Mg, Mn, Zn) to explore the role of electronic factors involved in this adsorption process, where Mg(dobpdc) shows reversible adsorption and Zn/Mn (dobpdc) show reversible CS₂ adsorption.

While simulations may enable quantification of individual electronic factors in a future study, we believe we have provided ample evidence to explain the observed trends and metal-to-metal differences in the adsorption process in the present manuscript. The distinct reversibility observed for mm-2-Mg₂(dobpdc) is the result of the appreciably lower thiophilicity of Mg in comparison with Mn and Zn, as

has been well established in the literature and discussed in the manuscript⁴. For desorption (i.e., reversibility) to occur, the coordinating dithiocarbamate species must be removed from the metal center through the breaking of the M–S bond, and thus greater thiophilicity inevitably increases the thermodynamic barrier to this bond scission. Indeed, dithiocarbamates are a broadly used ligand in coordination chemistry, and the low reversibility of their coordination to first-row transition metals resulting from their high thiophilicity has been broadly demonstrated⁵. As such, the irreversibility of adsorption on mm-2–Mn₂(dobpdc) and mm-2–Zn₂(dobpdc) is not surprising. In contrast, the Mg–S bond formed upon CS₂ adsorption in mm-2–Mg₂(dobpdc) can be broken with mild thermal input owing to the poor thiophilicity of Mg. We have added further emphasis on these points to the manuscript for clarification.

Reviewer #3:

We thank the reviewer for the careful reading and helpful comments.

The Reviewer's primary misgiving is a belief that the reported cooperative adsorption mechanism is effectively derivative of pore-gating/breathing and/or intrapore condensation. As we discuss in response to specific comments below, the reactive cooperative adsorption mechanism described in this manuscript is fundamentally distinct and provides critical advantages not found in pore-gating mechanisms. Additionally, we emphasize that this work is intended as a fundamental exploration of the broader utility of this new cooperative chemisorption mechanism through the study of an exotic, relatively unexplored adsorbate. While we highlight aspects in which this reactivity manifests as desirable functional properties, we do not intend to propose immediate incorporation of this material into an industrial process.

The Reviewer mentions a lack of new characterization techniques as a reservation toward the publication of this work. While we commend any publication that offers a new avenue to understand the complex interactions governing gas storage and separations, we believe the battery of characterization techniques employed in this work was more than sufficient to achieve our aims in elucidating a new example of chemisorptive cooperativity. Indeed, many of the reported techniques required exceptional expertise with a broad range of highly specialized instrumentation, often with first-of-a-kind modifications to incorporate toxic CS₂ vapor. The following techniques used in this study undoubtedly set a high bar for characterization: synchrotron PXRD of vapor-dosed samples, *in situ* XAS using a custom gas cell combined with DFT and molecular dynamics calculations, synchrotron SCXRD studying single-crystal-to-single-crystal transformations, and *in situ* DRIFTS using a custom-built instrument.

Below we respond in detail to each of the Reviewer's comments and describe our resulting efforts to improve the manuscript and clarify some nuanced but critical details that we believe make this study a particularly exciting contribution to the porous materials community.

Figure 1 is a typical textbook representation of adsorption isotherms and the importance of working capacity. I am not sure if this should be included, but now that it is included, the representation of the working capacity in a temperature swing process is wrong. If the process is isobaric, $P_{\text{ads}} = P_{\text{des}}$. Of course, it can be argued that if the adsorber chamber is isolated and closed, heating up will imply an increase of the pressure... but this is not the idea, as it will not allow adsorbent regeneration. As the authors do correctly later on, they obtained the working capacity in an isobaric process, where the working capacity is higher – I have some questions though, as the relative pressure could have gone to zero.

This figure is intended to demonstrate the ability of materials with step-shaped isotherms to achieve large working capacities with minimal temperature swings. Although readers with knowledge of gas adsorption may find this figure intuitive, we believe that those less familiar with the field of gas adsorption, such as the general audience of *Nature Communications*, will benefit from the idealized schematic. While many potential cycle configurations are possible, we chose to depict $P_{\text{des}} > P_{\text{ads}}$ for an idealized temperature-swing process in which recovery of high-purity CS₂ is desirable. In this case, in the absence of a purge gas, heating the bed will result in enrichment of the vapor phase with CS₂ as it desorbs from the framework. This is a common assumption for temperature-swing adsorption processes involving recovery of the heavy component. For example, see *Faraday Discuss.* **2016**, 192, 153, in which a process model was developed for TSA with diamine appended-frameworks for CO₂ capture (adsorption of CO₂

from coal flue gas with $P_{\text{ads}} = 0.12$ bar; desorption at elevated temperature under 1 bar CO_2)⁹. A similar schematic for Langmuir isotherms also appears in *Nat. Mater.* **2012**, *11*, 633 (Fig. 4)¹⁰.

The primary goal of the described cycling experiment is twofold: (i) evaluating the integrity of mm-2-Mg₂(dobpdc) through multiple adsorption–desorption cycles, which was non-obvious given the chemical adsorption mechanism and irreversibility for Mn and Zn, and (ii) demonstrating that the thermal sensitivity of the adsorption threshold pressure allows for the recovery of nearly the full adsorption capacity with mild thermal input. This first goal is currently stated in the manuscript, and we have added a point of emphasis on this second goal. As this is a fundamental study of the adsorption process, and empirically mimicking the dynamic industrial cycling process for high purity CS_2 recovery (Figure 1b) is outside the scope of the current work, we sought to demonstrate adsorption–desorption cycling and material integrity using equilibrium adsorption measurements under a controlled environment with a single changing variable, i.e., temperature. During this experiment, the CS_2 pressure was equilibrated at 75 mbar for each point at the respective temperature. With reference to the idealized process represented in Figure 1b, we have included a calculated working capacity for desorption under the highest CS_2 pressure measured and elevated temperature in the manuscript, but we wish to re-emphasize that we are constrained to a relatively narrow experimentally accessible window and that this experiment is not meant to model a specific process.

Page 3. “metalorganic frameworks that bind small molecules in a cooperative fashion show minimal guest uptake below a temperature dependent threshold pressure, followed by a sudden and sharp rise, or step, in adsorption (a Type V adsorption isotherm; Fig. 1b)8,9”. Reference 9 is a completely different behavior to what the authors include in this sentence, as this is not cooperative adsorption as such, but rather a breathing mechanism found in soft porous materials such as MIL53. In any case, I agree the shape of the isotherm is similar – but this is my point, this adsorption performance (shape of the isotherms) is not new and has been described many times in the past 1020 years.

We agree with the Reviewer that reference 9 was an incorrect choice, adding unintended ambiguity. This reference has been replaced with *Nature*, **2017**, *550*, 96 in which step-shaped chemical adsorption of carbon monoxide at open iron sites arises from site-to-site communication through a spin transition mechanism¹¹. Similar to the insertion mechanism discussed herein, this process yields step-shaped isotherms through a chemically specific adsorption process.

Page 4. “we recently discovered the first example of a material displaying cooperative adsorption via a unique, chainforming mechanism8”. The paper was published in Nature – fair enough. However, this kind of behavior/performance is not as novel as the authors are suggesting, and is indeed similar to many gate opening mechanisms reported in the last 10-20 years in the field of porous coordination polymers (breathing, gateopening). Of course, the adsorption mechanism is not the same, but the performance due to the reorganization of amines (or other organic entities in PCPs) is not unique nor novel. Hill coefficients have indeed been used for adsorption isotherms long time ago – even outside of the MOF community (see for example the work from Kanoh and Kaneko). The shape of the isotherms is common enough to be included in the IUPAC classification since the last century – I would not define this unique and unprecedented.

In this comment and the previous response, the Reviewer mentions that the step-like isotherms of the reported material appear similar to those of materials with adsorption proceeding through breathing/flexibility or intrapore condensation. Here we wish to draw critical distinctions between these classes of step-like isotherms in terms of both the underlying chemistry and the ultimate utility. First, the cooperative chemical adsorption process herein is not simply a product of the “reorganization of amines”. This is a chemical reaction on the surface of the framework, in which the adsorbate inserts into the metal–amine bond, coinciding with proton transfer to produce new chemical functionalities (i.e., dithiocarbamate and ammonium) that can interact with one another in 1D along the pores of the framework. This adsorptive process is distinct from previous works in which dangling organic moieties were used to block the pores, of either rigid or flexible frameworks, in a purely physical adsorption process. Second, frameworks relying on breathing or flexibility typically undergo large volume changes upon adsorption and desorption, creating considerable engineering challenges in process design¹². In contrast, the cooperative chemical adsorption process reported here occurs through a surface reaction along a relatively rigid pore, resulting in minimal volume change upon adsorption/desorption. Third, this reactive process is chemically specific, in that only certain molecules with particular properties, as discussed in the introduction of the main text, can be cooperatively adsorbed in this fashion. In contrast, in pore-gated materials, many chemicals are capable of being adsorbed in a step-shaped fashion, often with minimal molecular prerequisite for inducing framework breathing.

Noting these distinctions, we believe that broadening the utility of this desirable and unique cooperative chemical adsorption mechanism to a new adsorbate will be of interest to the general readership of *Nature Communications*. We recognize the important precedent set by many other groups in achieving step-shaped isotherms over the past decade, but we emphasize the significant fundamental advance afforded by the design of new materials with step-like isotherms resulting from reversible chemisorption.

Page 4: “Despite the obvious practical advantages, thus far all investigations of this cooperative process have been limited to CO₂ adsorption^{8,10,11}” I completely disagree with this statement. A typical example about MOFs on this are the recent works on water adsorption from e.g. Yaghi et al. What is true is that cooperative interaction typically takes place with molecules with high intermolecular interaction – so it is more difficult to happen with e.g. methane at room temperature, although it has been described for this gas molecule at much lower temperatures (eg 125K, Yildirim et al.) or by the same authors of the current manuscript three years ago using, again, a flexible structure.

From the context of the introduction, we believe it is evident that this point is a direct reference to the cooperative insertion mechanism previously elucidated by our group for CO₂ in diamine-appended M₂(dobpdc)^{1,2}. To date, this reactive cooperative mechanism has only been demonstrated for CO₂. The previous studies mentioned by the reviewer relate to fundamentally different adsorption processes, such as flexibility or intrapore condensation. For our materials, the driving force for cooperative adsorption is not guest–guest interactions, but rather host–guest reactivity. To ensure the clarity of our claim, we have revised the wording to specify a “cooperative **chemical** adsorption process”

Page 5. “Microcrystalline powders of M₂(dobpdc) (M = Mg, Mn, Zn) were prepared and characterized according to previously published procedures”. Measuring Langmuir surface area for a microporous material in this century is simply wrong. This was done broadly by the MOF community when these

materials were discovered, but the results are not adequate since they are only valid to give higher values for the specific surface area. This is well described in the recommendations from the IUPAC, also for MOFs. The N₂ adsorption isotherms reported in the supplementary information are extremely far away from the quality expected for an academic journal. In MOF74like MOFs, the adsorption takes place at very low relative pressures, and this is where the data needs to be compared (as is done for all the other gases reported). The reported data gives information about the pore volume, but there is no comparison on the very low pressure range (0.01p/p₀), or the pore size distribution. More importantly, if N₂ saturation takes place at ca 760 mbar, how do the authors reach 1000 mbar? The authors have a 3Flex available that they should use always.

We recognize the importance of reporting BET surface areas for all newly discovered metal–organic frameworks, and indeed, our group previously reported BET surface areas for the bare framework materials^{2,13}. Measuring Langmuir surface areas with low data point density (such as those initially included in the supporting information) provides a rapid means of verifying the porosity of known frameworks. However, we agree with the reviewer that publishing such data can perpetuate bad practice in the community. As a result, we have measured new 77 K N₂ and 87 K Ar adsorption data for M₂(dobpdc) (M = Mg, Mn, Zn) with high data density at low pressures and added this to the supporting information, along with the resulting BET surface areas and pore size distribution analysis.

The reviewer appears to have confused the mbar units for these plots with mmHg or torr. N₂ saturation at 77 K occurs around 1013 mbar, or 760 mmHg (torr).

Page 6 “Using linear spline fits and the Clausius–Clapeyron relationship”. Linear spline is a technique for interpolation rather than fitting. In this case, looking at eg Sup. Figure 23, there is some obvious noise in the jump due to equilibration issues, but there is no fitting here but an interpolation. Of course, a proper fitting needs to be done (there are multiple equations in the literature), and the Clausius–Clapeyron representation should be included in the ESI.

We have revised the wording in the manuscript and supporting information to reflect that the spline method is an interpolation technique rather than a true isotherm fit. We agree with the Reviewer that in most cases, a physically meaningful isotherm model is superior to interpolation. However, for these frameworks, adsorption proceeds through a complex mechanism involving both chemical reaction and condensation. Rather than relying on the assumptions of a known isotherm model, which may not capture the nuances of this system, we chose to favor the raw data by using interpolation to calculate the adsorption enthalpies. Very long equilibration times (240 s, with equilibration defined as <0.01% change in average pressure during that interval) were allotted to minimize noise in the data; however, some degree of noise remained unavoidable. Regardless, the data point density is sufficiently high and the noise sufficiently low to allow calculation of the adsorption enthalpy at the reported loading of 2 mmol/g (in the step region) using spline interpolation. We believe this limited application of interpolation is appropriate in this case. As a consistency check, the adsorption enthalpy of CS₂ in the bare framework in the condensation region (*n* = 12 mmol/g) was calculated as -27 ± 2 kJ/mol using linear spline interpolation—this agrees well with the anticipated $\Delta_{\text{vap}}H^\circ$ for CS₂ of 27.5 ± 0.6 kJ/mol (NIST WebBook).

We have added a plot of $\ln P$ vs. $1/T$ at a loading of 2 mmol/g for mm-2–Mg₂(dobpdc) to the supporting information as a representative demonstration of the application of the Clausius–Clapeyron relationship.

“When CS₂ adsorption was measured for Mg₂(dobpdc) appended with isopentylamine, a primary monoamine with a similar steric profile to mm2, only Langmuir type adsorption was observed with a Δh_{ads} of just -26.0 ± 0.3 kJ/mol (Supplementary Fig. 26).” Again, these isotherms are not Type I since they show some steps – this can be demonstrated by trying the fitting of a Langmuir equation instead of the spline interpolation included in Fig. S27.

Thank you for the helpful comment. We have changed the text of the manuscript to reflect this point. “When CS₂ adsorption was measured for Mg₂(dobpdc) appended with isopentylamine, a primary monoamine with a similar steric profile to mm-2, similar step-shaped adsorption was not observed and a Δh_{ads} of just -26.0 ± 0.3 kJ/mol was calculated.”

“Impressively, individual isotherms of various gases at 75 °C, including CO₂, suggest a near-perfect selectivity for CS₂ (Fig. 3b).” This is true and expected for molecules smaller and with lower guest-guest interactions than CS₂, but not for water. Why was the isotherm measured up to 30 mbar only and not 1000 mbar as the others?

For water isotherms, the maximum pressure is limited by the saturation pressure at the coldest point in the instrument. As a result, we are restricted to a maximum pressure of about 30 mbar ($P/P_0 = 0.08$) at 75 °C to avoid condensation in areas of the instrument that cannot be heated. Of note, we have previously reported a full set of water isotherms for mm-2–Mg₂(dobpdc),¹ where the adsorption behavior closer to saturation can be observed in the lower-temperature isotherms.

We emphasize that the mechanisms for water adsorption and CS₂ adsorption are different in this material: while water adsorption depends primarily on guest–guest interactions (as mentioned by the Reviewer), CS₂ adsorption instead occurs primarily through host–guest reactivity in an insertion reaction to yield sulfur-bound ammonium dithiocarbamate chains. Water cannot participate in this mechanism. We have rephrased this sentence in the text to re-focus on the chemical specificity of the insertion process.

Figure 3. Why is the isotherm for mm2–Mg₂(dobpdc) at 25C different in Fig. 3a and Fig. 3c? At 100 mbar one reaches 8 mmol/g but the other only 4 mmol/g.

Also, for Fig. 3c there is a point of the isotherm (6 mbar) significantly displaced. Also, the desorption isotherms are only provided for 25C – this should be done for the whole range of temperatures or, at least, for 100C (MgMOF), to see how reversible the equilibrium is. This would help to clarify the methods employed in the analysis during the temperature swing adsorption process. For how long is the reactivation done? Why is the CO₂ included during the cooling down instead of using the N₂ flow? Also, if I understood correctly, these are not isobars, because there is no CS₂ during the heating, so the relative pressure is zero – is this correct?

Adsorption capacity in Figure 3a is in mmol/g and Figure 3c is in mol/mol, as shown in the axis titles in the graphs themselves and remphasized in the figure caption. Figure 3c is plotted as mol/mol rather than mmol/g to adjust for the considerable difference in molecular weight between Mg, Mn, and Zn₂(dobpdc), allowing for more direct comparison of the three frameworks. An analogous plot to Figure

3c with adsorption capacity in mmol/g, is included in the supporting information (Supplementary Figure 29).

The position of the point at 6 mbar was repeatedly observed for this measurement with the very long reported equilibration time of 240 seconds. During the collection of this data it was observed that with shorter equilibrium times this displacement was significantly greater. As we gradually increased the equilibration time, we reached a time (i.e., 240 seconds) where little change in this point was observed with increasing time. Additionally, with the 240 second equilibrium period, each isotherm takes over two weeks to collect. It should be noted that the x -axis in Figure 3c is logarithmic, accentuating this displacement.

Desorption data for all temperatures is included in the supporting information (Supplementary Figure 24). Of note, full desorption is observed at 100 °C with little hysteresis.

In the cycling experiment, the sample was held at 100 °C for 30 minutes to allow for temperature equilibration to occur before equilibrium adsorption data at 75 mbar were collected. We have added this detail to the figure caption (Figure 3) and supporting information.

The Reviewer may have misinterpreted how and why the isobaric TGA experiments were run. We have attempted to clarify this point with additional detail in the manuscript. At this point in the manuscript, we had shown from FT-IR that while mm-2-Zn₂(dobpdc) irreversibly chemically adsorbs CS₂, the Mn and Mg analogs appear to lose CS₂ upon heating under vacuum or N₂ flow. Thus, it was important to determine whether or not the loss of CS₂ vibrations in the FT-IR spectra correlated to clean reversibility of the adsorbent, returning it to its functional, diamine-bound state. We know that these materials adsorb CO₂ in a cooperative manner (as has been established previously¹ and in the supporting information herein). We have widely shown that TGA-based isobaric CO₂ adsorption is a reliable and rapid method to characterize CO₂ adsorption in these materials^{1,2,8}, and CS₂ equilibrium adsorption measurements are significantly more difficult to run and time-consuming. As a result, we used TGA-based isobaric CO₂ adsorption as the initial tool to characterize functional reversibility of CS₂ adsorption. In these experiments, CS₂-exposed mm-2-Mn₂(dobpdc) and mm-2-Mg₂(dobpdc) are heated under flowing N₂ to remove the adsorbed CS₂ and then cooled under flowing CO₂ to measure the ability of the material to cooperatively adsorb the target gas after exposure to CS₂ and subsequent reactivation. In these experiments CO₂ is the target gas being adsorbed, not CS₂. We are unfortunately unable to perform such flow experiments with CS₂ with our instrumentation, as CS₂ is a flammable and toxic vapor. The details of how these experiments were done and full plots of these experimental data, with time stamps, can be found in the original supporting information. The information gathered from these experiments, namely the functional reversibility of mm-2-Mg₂(dobpdc), justified further characterization of the Mg analog through equilibrium temperature-swing adsorption cycling with CS₂.

Finally, I found the 6.8 mmol/g working capacity of mm2Mg2(dobpdc) interesting. The process takes advantage of the s-shape of the isotherms. However, the presence of the mm in the porosity also limits the pore volume and adsorption capacity. In this regard, the authors do not mention an easier alternative they have in their results: nonmodified Mg2(dobpdc) shows a capacity of ca. 19 mmol/g at 25C and 75mbar, and an loading of ca 6 mmol/g at 75C and 75 mbar (I suspect much lower at 100C, say 4mmol/g) – the isotherms being fully reversible at these temperatures. This translates to working capacities of 13-15

mmol/g, twice or three times higher than the ones reported in the modified MOFs and without any issues on reversibility. I am wondering if I missed something important here.

Though the observation of a higher theoretical working capacity in $\text{Mg}_2(\text{dobpdc})$ under isobaric (75 mbar) cycling conditions is valid, the working capacity is actually lower if following the adsorption–desorption cycle demonstrated in Figure 1b, specifically desorption under higher pressures of CS_2 in order to achieve a high purity CS_2 product stream. If we consider a process for the bare framework involving adsorption at 75 mbar and 25 °C and desorption at 300 mbar and 75 °C, a working capacity of ~3 mmol/g is obtained (possibly up to 4–5 mmol/g if desorption is instead performed at 100 °C). As is now discussed in the manuscript, for a similar process mm-2– $\text{Mg}_2(\text{dobpdc})$ would have a working capacity of 6.3 mmol/g. 300 mbar is used for this comparison as we are constrained to a relatively narrow experimentally accessible window. Also, where humidity would complicate adsorption in the bare framework through competitive adsorption at the open metal sites, as water can not undergo the cooperative chemical adsorption pathway, it will not interfere with reactive CS_2 adsorption in the mm-2– $\text{Mg}_2(\text{dobpdc})$. This is evidenced by there being no appreciable difference in FT–IR band position when CS_2 exposure of mm-2– $\text{Mg}_2(\text{dobpdc})$ was conducted using a gas adsorption instrument or through vapor exposure in ambient atmosphere.

The lowered effective porosity of mm-2– $\text{Mg}_2(\text{dobpdc})$ is not a concern, as this is a reactive adsorption process. For chemically specific adsorption of CS_2 (or CO_2) there only needs to be enough porosity to facilitate diffusion of 1 CS_2 per diamine, which is evident to exist. As discussed in the manuscript, from the observed adsorption capacities in mm-2– $\text{Mg}_2(\text{dobpdc})$ relative to the expected loading for 1 CS_2 per diamine and the capacity remaining upon exposure to vacuum at 25 °C it is clear that additional physical adsorption occurs upon the chemical adsorption pathway. For mm-2– $\text{Mg}_2(\text{dobpdc})$ at 25 °C the capacity is ~50:50 chemical adsorption and physical adsorption.

We wish to reemphasize, as stated in response to the first comment, that the shown cycling experiment is meant to show access the near full step capacity in a mild temperature swing process and that the integrity of the material is maintained during adsorption and subsequent desorption, and that this experiment is not meant to model a specific process.

References

- 1) Siegelman, R. L. *et al.* Controlling cooperative CO_2 adsorption in diamine-appended $\text{Mg}_2(\text{dobpdc})$ metal-organic frameworks. *J. Am. Chem. Soc.* **139**, 10526-10538 (2017).
- 2) McDonald, T. M. *et al.* Cooperative insertion of CO_2 in diamine-appended metal-organic frameworks. *Nature* **519**, 303-308 (2015).
- 3) Irving, H. & Williams, R. J. P. The stability of transition metal complexes. *J. Chem. Soc.* **537**, 3192-3210 (1953).
- 4) Kepp, K. P. A quantitative scale of oxophilicity and thiophilicity. *Inorg. Chem.* **55**, 9461-9470 (2016).
- 5) Hogarth, G. in *Progress in Inorganic Chemistry, Vol. 53* (ed Kenneth D. Karlin) Transition Metal Dithiocarbamates: 1978-2003, 71-561 (John Wiley & Sons, Inc. , 2005).
- 6) Mason, J. A., Sumida, K., Herm, Z. R., Krishna, R. & Long, J. R. Evaluating metal–organic frameworks for post-combustion carbon dioxide capture via temperature swing adsorption. *Energy Environ. Sci.* **4**, 3030 (2011).

- 7) Drage, T. C. *et al.* Materials challenges for the development of solid sorbents for post-combustion carbon capture. *J. Mater. Chem.* **22**, 2815-2823 (2012).
- 8) Milner, P. J. *et al.* A diaminopropane-appended metal-organic framework enabling efficient CO₂ capture from coal flue gas via a mixed adsorption mechanism. *J. Am. Chem. Soc.* **139**, 13541-13553 (2017).
- 9) Hefti, M., Joss, L., Bjelobrk, Z. & Mazzotti, M. On the potential of phase-change adsorbents for CO₂ capture by temperature swing adsorption. *Faraday Discuss.* **192**, 153-179 (2016).
- 10) Lin, L. C. *et al.* In silico screening of carbon-capture materials. *Nat. Mater.* **11**, 633-641 (2012).
- 11) Reed, D. A. *et al.* A spin transition mechanism for cooperative adsorption in metal-organic frameworks. *Nature* **550**, 96-100 (2017).
- 12) Mason, J. A. *et al.* Methane storage in flexible metal-organic frameworks with intrinsic thermal management. *Nature* **527**, 357-361 (2015).
- 13) Gygi, D. *et al.* Hydrogen storage in the expanded pore metal-organic frameworks M₂(dobpdc) (M = Mg, Mn, Fe, Co, Ni, Zn). *Chem. Mater.* **28**, 1128-1138 (2016).

Reviewers' comments:

Reviewer #1 (Remarks to the Author):

The authors have addressed my main concerns and I can recommend for publication. Two small things to check:

1. Figure S28, the 25C isotherm seems to be slightly lower than the 75 degree one at $P/P_0=0.01$. This is different to figure 3a. Which one is correct?
2. Since the water vapour sorption was only measured to $P/P_0=0.08$; this needs to be noted in the main text to avoid further confusion. i.e. at higher P/P_0 , the samples do adsorb water, it is subject to the instrument and such data was not obtained.

Reviewer #2 (Remarks to the Author):

This manuscript has been revised properly and could be accepted as it is.

Reviewer #3 (Remarks to the Author):

Please see the attached file.

I do not think the authors understood the points I raised, and I am sorry for the misunderstanding. First of all, I never assumed a pore-gating or any flexibility in this adsorption mechanism. I said the performance was similar but that the mechanism was different. This is both valid when I discuss the adsorption of CS₂ or water, but as I said, I mainly focus on performance.

I also mentioned in my previous reply the vast amount of techniques used by the authors. I do not think a paper in Nat. Commun. needs to include a new technique per se - I simply do not think the field and techniques is that new as the authors claim with multiple e.g. "first-of-a-kind", "unique", "unprecedented", "*Impressively*", statements, and that the general tone of the paper should be more humble, especially now that has capture the attention of the editors. Indeed, I fully agree with fact that the techniques used in the manuscript require a lot of expertise and that this is a very good piece of work. Having said that, the main difference I can see between this work and previous ones from the authors is the use of CS₂.

I understand the authors want to highlight the interesting adsorption mechanism found in the past, and now regarding CS₂, instead of focussing on finding and studying a specific process in terms of cycling, temperatures and pressures. The authors say "we wish to re-emphasize that we are constrained to a relatively narrow experimentally accessible window and that this experiment is not meant to model a specific process". Fair enough from a fundamental point of view - but this scheme from Figure 1 does not represent an optimal process. The criticism here is that taking $P_{des} > P_{ads}$ means that a Type I adsorption mechanism (or similar) performance is much worse than the Type V. Having said that, I fully agree (as many others in the past) that the step-based isotherms are an ideal situation for regeneration - I just think we need to be fair for comparison between Type I and Type V isotherms. In particular here, although Figure 1 seems to be a naively way of representing temperature swing adsorption for the benefit of the general audience of Nature Communications, it also introduces a source of misinterpretation of the results that needs to be clarified before getting accepted. The discussion can be summarized here in these two points:

1) Using a $P_{des} > P_{ads}$ is not optimal in a real process. In their reply the authors say "demonstrated in Figure 1b", but this is just a representation, not a proof, and as I said above, this process is not optimal. There is no need to include a flow of gas to remove CS₂ (although I am pretty sure there are some very smart solutions out there, this is not the point of this paper), just simply allowing the gas to expand will be much more efficient - so P_{des} does not need to be $< P_{ads}$ (optimal), but equal.

2) Water adsorption is true is possible in an empty mg₂(dobpdc), but the previous work from the same authors (see J. Am. Chem. Soc. 2015, 137, 4787–4803) show a high slope in the isotherm of water in mm-2-mg₂(dobpdc) at 25C, and I guess higher loadings. Again, a different mechanism to CS₂, but this is not a hydrophobic, functionalized MOF.

Some of the other referees made some questions about the impact of occupying part of the volume of the porosity by the amine groups. I think the data presented in the supporting information shows that this is a critical issue here. I am trying to represent this here:

Where (left) is mg2(dobpdc) and (right) is mm-2-mg2(dobpdc), taken from the main document and SI. For mg2(dobpdc), the working capacity is much higher if $P_{des} = P_{ads}$. Why would one try to desorb a gas using higher pressure? Just allow the gas to expand, so the performance increases. From a real applied point of view (out of the scope here) both situations will not be optimal because the kinetics will be too slow – there is no fast way to push the gas into a pipe, but again there are smart ideas out there. Now, another difference between (left) and (right) is the y-axis scale. If we use a similar one, the results will be:

This is showing the impact of the functionalization in the pore volume and working capacity. Simply, the CS₂ loading will never be as large as in the original material. The best way to solve this discussion is to include some breakthrough experiments similar to the ones that the authors have included in the past (see J. Am. Chem. Soc. 2015, 137, 4787–4803) including CS₂ in the presence of moisture and to compare the performance of both mg2(dobpdc) and mm-2-mg2(dobpdc). This is much simpler than the other characterization techniques included here.

In addition to this, there were some points raised during the discussion:

- Page 10. “We recognize the important precedent set by many other groups in achieving step-shaped isotherms over the past decade (...)”. I refer here to my first comments in my previous reply. I would like to see then this recognition in step-shaped isotherms (again, I agree the mechanism is different) - having a more humble (and realistic) approach to the discussion of the results will be very welcome.

- “The previous studies mentioned by the reviewer relate to fundamentally different adsorption processes, such as flexibility or intrapore condensation. For our materials, the driving force for cooperative adsorption is not guest–guest interactions, but rather host–guest reactivity”. What I think the authors did not understand is the fact that, even if from a fundamental point of view a mechanism is different, the shape of the isotherm will govern the performance of a material. I want to see this discussion reflected in the revised version.

- Page 11. Regarding the N₂ adsorption isotherms, I apologise for the confusion between mbar and torr - my fault. I thank the authors for re-measuring the N₂ isotherms, but I still see some of the old ones in the Supp info (e.g. Supp Figs. 8, 17, 21, etc.). Please update them.

- Fitting of isotherms. My point was that any fitting would be much more desirable than interpolation, even without a proper adsorption model. See for example Supp. Fig. 26 and the interpolation done at 50C; also the isotherm at 25C has some issues (this is better observed in Supp. Fig 28). This will not take too much time from the authors.

- Page 12. “Of note, we have previously reported a full set of water isotherms for mm-2–Mg₂(dobpdc),¹ where the adsorption behaviour closer to saturation can be observed in the lower-temperature isotherms.” I was not able to find the water isotherm in the reference [1] but in J. Am. Chem. Soc. 2015, 137, 4787–4803 the material seems to adsorb a substantial amount of water. I agree water cannot participate in the same mechanism, but this does not mean the material does not adsorb lots of water, making a big impact in the performance (again, my previous comments between fundamental mechanism and performance). **Please, include full water adsorption isotherm for mm-2–Mg₂(dobpdc) at (minimum) 25C and 75C – again this is simpler than most of the characterization techniques included here.** I see referee #1 has some similar concerns.

Response to Reviewers:

Manuscript ID: NCOMMs-18-12050-T

Title: “Cooperative Adsorption of Carbon Disulfide in Diamine-Appended Metal–Organic Frameworks”

Authors: C. Michael McGuirk, Rebecca L. Siegelman, Walter S. Drisdell, Tomče Runčevski, Phillip J. Milner, Julia Oktawiec, Liwen F. Wan, Gregory M. Su, Henry Z. H. Jiang, Douglas A. Reed, Miguel I. Gonzalez, David Prendergast, Jeffrey R. Long

Reviewer #1:

We thank the reviewer for the careful reading of our responses and for the helpful comments that have made this a stronger manuscript.

1. Figure S28, the 25C isotherm seems to be slightly lower than the 75 degree one at $P/P_0=0.01$. This is different to figure 3a. Which one is correct?

Both plots are correct, as Figure S28 is in units of P/P_0 and Figure 3a is in units of absolute pressure.

2. Since the water vapour sorption was only measured to $P/P_0=0.08$; this needs to be noted in the main text to avoid further confusion. i.e. at higher P/P_0 , the samples do adsorb water, it is subject to the instrument and such data was not obtained.

Thank you for the comment. We have added this point to main text of the manuscript so that it is clear to the audience.

Reviewer #2:

We thank the reviewer for the careful reading of our responses and for their contribution to this manuscript.

Reviewer #3:

We thank the reviewer for the careful reading and helpful comments.

I do not think the authors understood the points I raised, and I am sorry for the misunderstanding. First of all, I never assumed a pore-gating or any flexibility in this adsorption mechanism. I said the performance was similar but that the mechanism was different. This is both valid when I discuss the adsorption of CS₂ or water, but as I said, I mainly focus on performance.

We believe that we did understand the Reviewer’s statements regarding comparing performance in different classes of adsorbents that yield Type V isotherms, and reemphasize that the adsorption profile, while obviously a huge factor, is not the sole consideration for performance and utility. In this vein, our previous response and respective edits to the manuscript detailed critical differences between the adsorption performance of cooperative chemical adsorbents and flexible or pore-gating frameworks. We additionally reiterate that this work is a fundamental exploration of the utility of a unique chemically reactive cooperative adsorption process, previously only observed to function for CO₂, and that there is no primary industrial application that performance in this manuscript is targeted towards.

To reiterate our previous response regarding some key differences between pore-gating/flexible frameworks and the adsorption process detailed in this manuscript that affect performance/utility: “First, the cooperative chemical adsorption process herein is not simply a product of the “reorganization of amines”. This is a chemical reaction on the surface of the framework, in which the adsorbate inserts into the metal–amine bond, coinciding with proton transfer to produce new chemical functionalities (i.e., dithiocarbamate and ammonium) that can interact with one another in 1D along the pores of the framework. This adsorptive process is distinct from previous works in which dangling organic moieties were used to block the pores, of either rigid or flexible frameworks, in a purely physical adsorption process. Second, frameworks relying on breathing or flexibility typically undergo large volume changes upon adsorption and desorption, creating considerable engineering challenges in process design¹. In contrast, the cooperative chemical adsorption process reported here occurs through a surface reaction along a relatively rigid pore, resulting in minimal volume change upon adsorption/desorption. Third, this reactive process is chemically specific, in that only certain molecules with particular properties, as discussed in the introduction of the main text, can be cooperatively adsorbed in this fashion. In contrast, in pore-gated materials, many chemicals are capable of being adsorbed in a step-shaped fashion, often with minimal molecular prerequisite for inducing framework breathing.”

I also mentioned in my previous reply the vast amount of techniques used by the authors. I do not think a paper in Nat. Commun. needs to include a new technique per se - I simply do not think the field and techniques is that new as the authors claim with multiple e.g. "first-of-a-kind", "unique", "unprecedented", "Impressively", statements, and that the general tone of the paper should be more humble, especially now that has capture the attention of the editors. Indeed, I fully agree with fact that the techniques used in the manuscript require a lot of expertise and that this is a very good piece of work. Having said that, the main difference I can see between this work and previous ones from the authors is the use of CS₂.

In the revised manuscript, we have removed many superlatives from the text and towards engendering a humbler tone.

The initial motivation of this work was the determination if the complex cooperative chemical adsorption process previously reported solely for CO₂ was indeed wholly specific to CO₂, or if some other small molecule could participate in this reactive adsorption mechanism. Through this broad initiative, we discovered that CS₂ could be cooperatively chemically adsorbed by diamine-appended M₂(dobpdc). Importantly, through our extensive characterization of this adsorption process we determined key differences between CO₂ and CS₂ adsorption behavior arising from their distinct chemical reactivities, as is fully detailed in the manuscript. Beyond the realization of an expanded utility for this adsorption mechanism and the detailing of observed differences in adsorption behavior between these two molecules (e.g., metal-dependent threshold pressures and reversibility), this work also provides fundamental insights into the reactivity and coordination chemistry of CS₂, as is particularly emphasized in the single-crystal XRD and conclusion section of manuscript, which is of considerable interest to the molecular coordination chemistry community, as dithiocarbamates are a prolifically studied ligating species. While a cursory assessment may conclude that the only difference is CO₂ versus CS₂, the work detailed in this manuscript undoubtedly supersedes this appraisal.

1) Using a Pdes > Pads is not optimal in a real process. In their reply the authors say "demonstrated in Figure 1b", but this is just a representation, not a proof, and as I said above, this process is not optimal. There is no need to include a flow of gas to remove CS₂ (although I am pretty sure there are some very smart solutions out there, this is not the point of this paper), just simply allowing the gas to expand will be much more efficient – so Pdes does not need to be < Pads (optimal), but equal.

We have changed Figure 1 to reflect adsorption and desorption occurring at the same pressure, and included calculated usable capacities for different isobaric TSA conditions in the main text.

2) *Water adsorption is true is possible in an empty mg₂(dobpdc), but the previous work from the same authors (see J. Am. Chem. Soc. 2015, 137, 4787–4803) show a high slope in the isotherm of water in mm-2-mg₂(dobpdc) at 25C, and I guess higher loadings. Again, a different mechanism to CS₂, but this is not a hydrophobic, functionalized MOF.*

mm-2–Mg₂(dobpdc) adsorption was *not* studied in this manuscript, but rather mmen–Mg₂(dobpdc), which is a secondary/secondary substituted diamine. Nonetheless, this study of CO₂ adsorption in the presence of water for various porous materials importantly highlights a key benefit of using chemically specific cooperative adsorption in diamine-appended Mg₂(dobpdc) over open-metal site frameworks, such as bare Mg₂(dobpdc) and Mg₂(dobdc). Namely, in diamine-appended Mg₂(dobpdc) the presence of high partial pressures of water does *not* interfere with the chemically specific cooperative adsorption of CO₂ (Figure 13, left, in that manuscript). In contrast, *the presence of water precludes almost all CO₂ adsorption in bare Mg₂(dobdc)* because the open metal sites of the framework are not chemically specific adsorption sites and are thus saturated with water (Figure 8 in that manuscript). We expect the same trend to hold true to CS₂ adsorption in the presence of water, where CS₂ adsorption by bare Mg₂(dobpdc) would be dramatically reduced in the presence of water or any other small molecule capable of coordination to an open metal site. Unfortunately, a similar mixed-gas study would be *extremely difficult, dangerous and time consuming* to perform on this specialized high-throughput instrument with highly toxic and corrosive CS₂.

Some of the other referees made some questions about the impact of occupying part of the volume of the porosity by the amine groups.

The other referee asked what would be the effect of lengthening the alkyl groups of mm-2, to which a satisfactory answer was given as they suggest publication of the current manuscript.

Where (left) is mg₂(dobpdc) and (right) is mm-2-mg₂(dobpdc), taken from the main document and SI. For mg₂(dobpdc), the working capacity is much higher if P_{des} = P_{ads}. Why would one try to desorb a gas using higher pressure? Just allow the gas to expand, so the performance increases. From a real applied point of view (out of the scope here) both situations will not be optimal because the kinetics will be too slow – there is no fast way to push the gas into a pipe, but again there are smart ideas out there. Now, another difference between (left) and (right) is the y-axis scale. If we use a similar one, the results will be: This is showing the impact of the functionalization in the pore volume and working capacity. Simply, the CS₂ loading will never be as large as in the original material.

The Reviewer is correct in that if we take 75 mbar as both the P_{des} and P_{ads} and directly compare the working capacities of bare Mg₂(dobpdc) and mm-2 Mg₂(dobpdc), as derived from the pure component adsorption isotherms, bare Mg₂(dobpdc) would have a higher capacity. Yet, the following points should be considered: (i) As previously stated there is no specific industrial process this manuscript is targeting, as such 75 mbar pressure is somewhat arbitrary. 75 mbar was chosen for the cycling experiments in the manuscript as it allowed us to study the reversibility of adsorption and robustness of mm-2 Mg₂(dobpdc) using a minimal temperature swing, while achieving optimally high adsorption capacity and removing hysteresis considerations. If we compare at a lower pressure, such as 10 mbar, and swing from 25 °C to

75 °C, we calculate a usable capacity of ~3.5 mmol/g for both systems, which includes hysteresis being taken into account. Also, the large capacity in the 75 mbar range for the bare framework results from intrapore condensation of CS₂. The location of these condensation steps would be very sensitive to co-adsorbents, making its use unreliable. (ii) As discussed in an earlier comment, *J. Am. Chem. Soc.* 2015, 137, 4787–4803 clearly demonstrates from mixed gas adsorption isotherms that, in the presence of water, adsorption at open-metal sites, such as those in Mg₂(dobpdc), is dramatically reduced due to water saturating the available adsorption sites. Whereas, adsorption via chemically reactive cooperative adsorption is maintained, as the adsorption mechanism is chemically specific. With this in mind, it is clear that the bare framework is not desirable, as the presence of other coordinative species in a real-world stream is inevitable and will immediately and dramatically reduce adsorption performance and thus utility. In contrast, this previous work supports that the presence of water will *not* have a detrimental effect on chemical CS₂ adsorption in mm-2–Mg₂(dobpdc). This is herein empirically supported by the fact that we do not observe a difference in the FT–IR spectra when CS₂ adsorption occurs in open atmosphere, as noted in the main text of the manuscript. (iii) *Importantly, as the Reviewer states, further discussion and consideration of real-world processes is beyond the scope of this work.*

The best way to solve this discussion is to include some breakthrough experiments similar to the ones that the authors have included in the past (see J. Am. Chem. Soc. 2015, 137, 4787–4803) including CS₂ in the presence of moisture and to compare the performance of both mg₂(dobpdc) and mm-2-mg₂(dobpdc). This is much simpler than the other characterization techniques included here.

We believe the reviewer has misinterpreted the work detailed *J. Am. Chem. Soc.* 2015, 137, 4787–4803. This manuscript does not use breakthrough, but rather equilibrium multi-component adsorption isotherms using a custom-built instrument. We agree that long term it would be informative to make adsorption measurements for CS₂ in the presence of controlled amounts of water vapor, but the highly toxic, flammable and corrosive nature of CS₂ precludes the performance of such measurements. To overcome the toxicity considerations, we would need to move this cumbersome adsorption instrument into a highly ventilated specialized walk-in hood. Additionally, this instrument is not designed to handle corrosive gases and we fear that the use of CS₂ would destroy this very expensive instrument.

The flow-based nature of breakthrough measurements yields even greater safety concerns, as well as inevitably being detrimental to our instrumentation. Such experiments would very difficult to perform, thus would be fitting for a stand-alone follow up study using specialized instrumentation.

- Page 10. *“We recognize the important precedent set by many other groups in achieving step-shaped isotherms over the past decade (...)”. I refer here to my first comments in my previous reply. I would like to see then this recognition in step-shaped isotherms (again, I agree the mechanism is different) - having a more humble (and realistic) approach to the discussion of the results will be very welcome.*

We have worked to remove superlatives from the main text to engender a humbler tone.

- *“The previous studies mentioned by the reviewer relate to fundamentally different adsorption processes, such as flexibility or intrapore condensation. For our materials, the driving force for cooperative adsorption is not guest–guest interactions, but rather host–guest reactivity”. What I think the authors did not understand is the fact that, even if from a fundamental point of view a mechanism is different, the shape of the isotherm will govern the performance of a material. I want to see this discussion reflected in the revised version.*

We have re-worked the introduction to clarify that there is strong precedent for adsorbents with Type V isotherms promoting more favorable adsorption–desorption cycling.

- Page 11. *Regarding the N₂ adsorption isotherms, I apologise for the confusion between mbar and torr – my fault. I thank the authors for re-measuring the N₂ isotherms, but I still see some of the old ones in the Supp info (e.g. Supp Figs. 8, 17, 21, etc.). Please update them.*

We have appropriately updated all 77 K N₂ isotherms in the supporting information.

Fitting of isotherms. My point was that any fitting would be much more desirable than interpolation, even without a proper adsorption model. See for example Supp. Fig. 26 and the interpolation done at 50C; also the isotherm at 25C has some issues (this is better observed in Supp. Fig 28). This will not take too much time from the authors.

Using the same CS₂ adsorption data for mm-2 Mg₂(dobpdc) that we performed linear spline interpolation with, we fit the *post-step* region of the isotherms to a mathematical model derived from the classical dual site Langmuir-Freundlich equation. In this manner, we calculated a differential enthalpy of adsorption (Δh_{ads}) at $n = 2$ mmol/g of 50 ± 6 kJ/mol. Importantly, this is within error of the value calculated using linear spline interpolation at the same loading (55 ± 5 kJ/mol). We have added this additional method to the supporting information. Similar fitting was performed with the isopentylamine–Mg₂(dobpdc) (dual site Langmuir-Freundlich) and bare Mg₂(dobpdc) (triple-site Langmuir-Freundlich) and is included in the supporting information. Fitting for both yields similar values to that calculated using linear spline interpolation.

- Page 12. *“Of note, we have previously reported a full set of water isotherms for mm-2–Mg₂(dobpdc), I where the adsorption behaviour closer to saturation can be observed in the lower-temperature isotherms.” I was not able to find the water isotherm in the reference [1] but in J. Am. Chem. Soc. 2015, 137, 4787–4803 the material seems to adsorb a substantial amount of water. I agree water cannot participate in the same mechanism, but this does not mean the material does not adsorb lots of water, making a big impact in the performance (again, my previous comments between fundamental mechanism and performance). Please, include full water adsorption isotherm for mm-2–Mg₂(dobpdc) at (minimum) 25C and 75C – again this is simpler than most of the characterization techniques included here. I see referee #1 has some similar concerns.*

See below for the water adsorption isotherms for mm-2–Mg₂(dobpdc) previously published in *J. Am. Chem. Soc. 2017, 139, 10526–10538*. This has been directly cited in the main text of the manuscript. Of note, all questions from other referees were fully satisfied with this data.

The material studied in *J. Am. Chem. Soc. 2015, 137, 4787–4803* is not the same as mm-2–Mg₂(dobpdc), but contains a secondary/secondary substituted diamine. Nonetheless, the important conclusion put forth by the manuscript the Reviewer is discussing (*J. Am. Chem. Soc. 2015, 137, 4787–4803*) is that although water can be co-adsorbed by diamine–appended Mg₂(dobpdc), this does **not** deter chemically reactive cooperative CO₂ adsorption (Figure 13). In contrast, CO₂ adsorption in open metal site frameworks, such as Mg₂(dobpdc) and Mg₂(dobdc), is effectively fully negated by the presence of water (Figure 8). Again, we expect this same phenomenon for CS₂ in the presence of water, but performing these experiments with CS₂ are not only outside the fundamental scope of this manuscript, but

also very dangerous. This previous work strongly supports the development use of a chemically specific cooperative adsorption process.

References

(1) Mason, J. A.; Oktawiec, J.; Taylor, M. K.; Hudson, M. R.; Rodriguez, J.; Bachman, J. E.; Gonzalez, M. I.; Cervellino, A.; Guagliardi, A.; Brown, C. M.; Llewellyn, P. L.; Masciocchi, N.; Long, J. R. Methane Storage in Flexible Metal-Organic Frameworks with Intrinsic Thermal Management. *Nature* **2015**, *527*, 357-361.

Reviewers' comments:

Reviewer #1 (Remarks to the Author):

1. I do not think that CS₂ is corrosive; toxic yes but so are lots of things (such as NH₃, NO, where breakthrough tests have been done under both dry and wet conditions). I do not think it is as difficult as the authors have described for a breakthrough with water, particularly if they can already manipulate CS₂ sufficiently via many techniques? At least, they could perhaps attempt some tests with MOF samples pre-loaded with water (i.e. using wet MOF samples).

2. If they can measure an isotherm with CS₂, and I suggest to compare that with the H₂O isotherms (as they reported in their previous papers) to give isotherm selectivity data, from IAST modelling for example.

3. I have a lot of sympathy with the view that CS₂ is just like CO₂ but heavier. Ref 3 states this and so it is important to state "What is the surprising result here?" to address his/her comments.

In general, I think the work can be published in Nature Commun after some revisions.

Response to Reviewers:

Manuscript ID: NCOMMs-18-12050-T

Title: "Cooperative Adsorption of Carbon Disulfide in Diamine-Appended Metal–Organic Frameworks"

Authors: C. Michael McGuirk, Rebecca L. Siegelman, Walter S. Drisdell, Tomče Runčevski, Phillip J. Milner, Julia Oktawiec, Liwen F. Wan, Gregory M. Su, Henry Z. H. Jiang, Douglas A. Reed, Miguel I. Gonzalez, David Prendergast, Jeffrey R. Long

Reviewer #1:

We thank the reviewer for the careful reading and helpful comments.

1. I do not think that CS₂ is corrosive; toxic yes but so are lots of things (such as NH₃, NO, where breakthrough tests have been done under both dry and wet conditions). I do not think it is as difficult as the authors have described for a breakthrough with water, particularly if they can already manipulate CS₂ sufficiently via many techniques? At least, they could perhaps attempt some tests with MOF samples pre-loaded with water (i.e. using wet MOF samples).

We find it important at this juncture to clearly state the questions at the core of this discussion. Namely: (i) does mm-2–Mg₂(dobpdc) chemically adsorb CS₂ in the presence of water through the mechanism discussed in the manuscript, and (ii) given the high capacity of CS₂ adsorption reported for bare Mg₂(dobpdc) in equilibrium adsorption measurements, why is adsorption in mm-2–Mg₂(dobpdc) advantageous?

In the vein of answering these questions, breakthrough experiments have been suggested. Yet, in addition to the many practical issues with performing breakthrough with this CS₂, it is our strong opinion that such experiments would not unambiguously yield the desired specific information, especially within the fundamental molecular perspective that is the focus of this paper. Moreover, such experiments would introduce considerable variables associated with targeting specific applications, which are well beyond the scope of this work, as previously discussed. Therefore, we have herein sought an alternative approach to addressing these questions, specifically through FT–IR spectroscopy, which has been shown in this manuscript to be a strong handle for probing adsorption in this system.

To address the first question, we studied the effects of prolonged water vapor exposure on CS₂ adsorption in mm-2–Mg₂(dobpdc), as evidenced by vibrations indicative of adsorption that have been characterized previously in this manuscript. The results are seen in the below figures. First, a control spectrum of activated mm-2–Mg₂(dobpdc) was collected (black trace). Second, activated mm-2–Mg₂(dobpdc) was dosed with CS₂ through vapor diffusion in a sealed vial for *10 minutes* (red trace). As expected, chemical adsorption is confirmed by dithiocarbamate vibrations around 960 and 1080 cm⁻¹, and physical adsorption is indicated by the vibration around 1520 cm⁻¹. Third, activated mm-2–Mg₂(dobpdc) was saturated with water through vapor diffusion in a sealed vial for *16 hours* (blue trace). The presence of water is clear from the broad vibrations from 3700–3100 cm⁻¹. Fourth, the water-saturated mm-2–Mg₂(dobpdc) was immediately dosed with CS₂ through vapor diffusion in a sealed vial for *10 minutes* (green trace). The chemical adsorption of CS₂ through dithiocarbamate formation in water-saturated mm-2–Mg₂(dobpdc) is unmistakably evidenced by vibrations around 960 and 1080 cm⁻¹. Additionally, the continued presence of water is clear from the broad vibrations from 3700–3100 cm⁻¹. As expected, the physical adsorption of CS₂ is deterred by the presence of water, as framework saturation with water would preclude the

necessary physical space for additional CS₂ beyond that which is chemically adsorbed. *This series of spectra unambiguously demonstrates that mm-2-Mg₂(dobpdc) is able to chemically adsorb CS₂ when fully saturated with water.* These spectra have been added to the supporting information and referenced in the body of the manuscript.

Supplementary Figure: Transmittance mode FT-IR Spectra of mm-2-Mg₂(dobpdc). Black line = activated, red line = CS₂ dosed for 10 minutes, blue line = H₂O dosed for 16 hours, and green line = H₂O dosed for 16 hours, then CS₂ dosed for 10 minutes. Dashed lines highlight bands arising from C-S vibrations in the CS₂ dosed sample.

To empirically address the second question, we studied the effects of prolonged exposure of bare Mg₂(dobpdc) to water vapor on CS₂ adsorption. The results are seen in the below figures. First, a control spectrum of activated Mg₂(dobpdc) was collected (black trace). Second, activated Mg₂(dobpdc) was dosed with CS₂ through vapor diffusion in a sealed vial for *10 minutes* (red trace). Physical CS₂ adsorption is evidenced by the vibration at ~1520 cm⁻¹. Third, activated Mg₂(dobpdc) was saturated with water through vapor diffusion in a sealed vial for *16 hours* (blue trace). The presence of water is clear from the broad vibrations from 3700–3100 cm⁻¹. Fourth, the water-saturated Mg₂(dobpdc) was immediately dosed with CS₂ through vapor diffusion in a sealed vial for *60 minutes* (green trace). Even after the much longer period of CS₂ exposure, it is evident from the very weak vibration at ~1520 cm⁻¹ that *minimal CS₂ is adsorbed in the water-saturated Mg₂(dobpdc)*. Owing to the strong adsorption of water in Mg₂(dobpdc), CS₂ adsorption is dramatically reduced. *It is clear from this result that the chemical promiscuity of adsorption in bare Mg₂(dobpdc) strongly detracts from the adsorption capacity seen in the highly environmentally controlled equilibrium adsorption measurements.* This result is directly in line with the previously referenced multi-component equilibrium adsorption experiments.¹ Indeed, in the presence of any other small molecule capable of being adsorbed, CS₂ adsorption capacity in bare Mg₂(dobpdc) will be dramatically reduced. *As such, the cooperative chemical adsorption mechanism of mm-2-Mg₂(dobpdc), which shows distinct **chemical specificity**, is clearly advantageous for the adsorption of CS₂ over bare Mg₂(dobpdc).* These spectra have been added to the supporting information and referenced in the body of the manuscript. In addition to these results, it should be noted that it has been previously demonstrated that bare Mg₂(dobpdc) demonstrates appreciable instability in ambient atmosphere.^{2,3} Moreover, functionalization of the open metal sites with a diamine has been demonstrated to stabilize the framework.⁴

Supplementary Figure: Transmittance mode FT-IR Spectra of Mg₂(dobpdc). Black line = activated, red line = CS₂ dosed for 10 minutes, blue line = H₂O dosed for 16 hours, and green line = H₂O dosed for 16 hours, then CS₂ dosed for 60 minutes. Dashed lines highlight bands arising from C-S vibrations in the CS₂ dosed sample.

2. If they can measure an isotherm with CS₂, and I suggest to compare that with the H₂O isotherms (as they reported in their previous papers) to give isotherm selectivity data, from IAST modelling for example.

Thank you for the suggestion. While IAST cannot be applied here because the mechanisms of water and CS₂ adsorption are fundamentally different, we can provide the noncompetitive selectivity, as done previously for comparing selectivity of sorbates adsorbed through different mechanisms in a given sorbent.⁴ Defined as $(q_{CS_2}/q_{H_2O})/(p_{CS_2}/p_{H_2O})$, the noncompetitive selectivity for CS₂ over water at 50 °C and 6 mbar is 22.4. The noncompetitive selectivity for CS₂ over water at 75 °C and 30 mbar is 11.1. We have included this detail in the manuscript.

3. I have a lot of sympathy with the view that CS₂ is just like CO₂ but heavier. Ref 3 states this and so it is important to state “What is the surprising result here?” to address his/her comments.

As the underlying adsorption mechanism explored in this manuscript is a chemical reaction, rather than physical adsorption, differences in chemical reactivity are highly consequential. As such, the discovery of any small molecule beyond CO₂ participating in this unique cooperative chemical adsorption mechanism that requires several specific chemical properties is quite surprising. Of note, other small molecules have been explored, but only CS₂ has been shown to be cooperatively and *reversibly* adsorbed. As emphasized in the manuscript, the study of CS₂ adsorption in synthetic porous materials has been effectively ignored to date. Thus, a study such as this, especially with the given adsorption mechanism, is quite unprecedented.

While CS₂ and CO₂ are structurally analogous, equating these two distinct chemicals based on this notion is reductive, as the differences in physical properties and chemical reactivity between CS₂ and CO₂ are innumerable. Similar logic could be applied to other structural homologs, such as H₂O and H₂S, but whereas we consume H₂O, H₂S is very poisonous, flammable and corrosive. Indeed, elemental substitution is not superficial. In this vein, several fundamental differences in CS₂ adsorption, from that of CO₂, were observed and are discussed throughout the manuscript. In particular, the extensively discussed consequences of varying the metal-center on step-threshold pressure and adsorption reversibility are highly reflective of the different chemical reactivity of

CS₂ and the coordination chemistry of dithiocarbamate ligands. Also in regards to the observed adsorption behavior, it is quite surprising to observe that additional physical adsorption occurs upon cooperative chemical adsorption, as this does not occur with CO₂ under similar conditions, which is reflected by the greater adsorption capacity of CS₂ relative to CO₂ in this sorbent. As we were surprised by this, we undertook extensive effort to characterize this phenomenon, as detailed in the DRIFTS section of the manuscript. We have added an extra point of emphasis regarding the uniqueness of this phenomenon in the manuscript.

Dithiocarbamates have been prolifically used as strong-field, irreversible *bidentate* ligands. Therefore, it was very surprising to observe, through the detailed suite of techniques, that the primary and secondary coordination environment in these frameworks work together to produce an *unprecedented dithiocarbamate coordination mode*. Without this exceptional coordination behavior, the observed reversibility of dithiocarbamate coordination would not occur, and this point is emphasized in the manuscript. Taken together, these points clearly demonstrate that although CS₂ and CO₂ are structurally similar, this work is highly unique, full of unexpected results, and broadly impactful.

References

- (1) Mason, J. A.; McDonald, T. M.; Bae, T. H.; Bachman, J. E.; Sumida, K.; Dutton, J. J.; Kaye, S. S.; Long, J. R. Application of a High-Throughput Analyzer in Evaluating Solid Adsorbents for Post-Combustion Carbon Capture Via Multicomponent Adsorption of CO₂, N₂, and H₂O. *J. Am. Chem. Soc.* **2015**, *137*, 4787-4803.
- (2) Vitillo, J. G.; Bordiga, S. Increasing the Stability of Mg₂(dobpdc) Metal–Organic Framework in Air through Solvent Removal. *Mater. Chem. Front.* **2017**, *1*, 444-448.
- (3) Vitillo, J. G.; Ricchiardi, G. Effect of Pore Size, Solvation, and Defectivity on the Perturbation of Adsorbates in MOFs: The Paradigmatic Mg₂(dobpdc) Case Study. *J. Phys. Chem. C.* **2017**, *121*, 22762-22772.
- (4) Milner, P. J.; Siegelman, R. L.; Forse, A. C.; Gonzalez, M. I.; Runcevski, T.; Martell, J. D.; Reimer, J. A.; Long, J. R. A Diaminopropane-Appended Metal-Organic Framework Enabling Efficient CO₂ Capture from Coal Flue Gas Via a Mixed Adsorption Mechanism. *J. Am. Chem. Soc.* **2017**, *139*, 13541-13553.

REVIEWERS' COMMENTS:

Reviewer #1 (Remarks to the Author):

The authors have addressed my concerns and I recommend the publication of this work.

Reviewer #3 (Remarks to the Author):

The comments from the authors on the flexibility are not necessary, as we all agree the phenomenon here is completely different to the breathing and gate-effect and is indeed similar to what the same laboratory has published in the last few years. The third point, though, "this reactive process is chemically specific, in that only certain molecules with particular properties, as discussed in the introduction of the main text, can be cooperatively adsorbed in this fashion" goes back to my previous comments on the selectivity in such material. The other big issue is the working capacity of the material proposed here.

A) Selectivity:

Although I 100% agree the proposed process is specific for certain molecules, my question is: Is this being strong enough to counteract for vdW, electrostatic and alternative physis/chemisorption processes. Is the proposed solution able to selectively bind CS₂ in humid conditions?

The authors argue that "in diamine-appended Mg₂(dobpdc) the presence of high partial pressures of water does not interfere with the chemically specific cooperative adsorption of CO₂". Again, I agree it will not follow the same "reactive process", but I will be surprised if the diamine groups are not also hydrophilic. With their results in hand, the fact that "we expect the same trend to hold true to CS₂ adsorption in the presence of water" **is only a hypothesis**. On one hand, a pure water isotherm will let know if the material is hydrophilic or not. On the other hand, a breakthrough curve for CS₂ adsorption in the presence of moisture, reproducing standard operating conditions, will be the definitive answer. CS₂ is toxic, but the authors have shown they can deal with it. If SH₂ produced in contact with water will attack the MOF, then this is another issue not included in the discussion.

B) Capacity:

Regarding the previous cartoon for the comparison of pressure: i) "there is no specific industrial process this manuscript is targeting, as such 75 mbar pressure is somewhat arbitrary". Working at non-optimal conditions, such as 10 mbar makes the bare framework to perform worse than at 75 mbar, but still comparable to the modified one. Why would the authors do that? Again, the comment about the fact that "the location of these condensation steps would be very sensitive to co-adsorbents, making its use unreliable" **is another hypothesis** not demonstrated in the answer. Simply check the optimal conditions for the bare and modified frameworks and compare them. My whole point here is that the functionalization has an impact on the pore volume, and this will decrease the working capacity of the material.

In order to be practical and solve this discussion, I ask the authors for mixture isotherms and a full water isotherm. The same safety issues that exist with CS₂ should be valid for humid conditions. In any case, the water isotherms provided are not full isotherms. Saturation pressure of water is 31.7

mbar at 298 K, so please do not stop at 0.15 p/p_0 , and make a systematic comparison of CS₂ and water adsorption in both materials, side by side.

Response to Reviewers:

Manuscript ID: NCOMMs-18-12050-T

Title: “Cooperative Adsorption of Carbon Disulfide in Diamine-Appended Metal–Organic Frameworks”

Authors: C. Michael McGuirk, Rebecca L. Siegelman, Walter S. Drisdell, Tomče Runčevski, Phillip J. Milner, Julia Oktawiec, Liwen F. Wan, Gregory M. Su, Henry Z. H. Jiang, Douglas A. Reed, Miguel I. Gonzalez, David Prendergast, Jeffrey R. Long

Reviewer #1:

The authors have addressed my concerns and I recommend the publication of this work.

We thank the reviewer for the careful reading and helpful comments.

Reviewer #3:

The comments from the authors on the flexibility are not necessary, as we all agree the phenomenon here is completely different to the breathing and gate-effect and is indeed similar to what the same laboratory has published in the last few years. The third point, though, “this reactive process is chemically specific, in that only certain molecules with particular properties, as discussed in the introduction of the main text, can be cooperatively adsorbed in this fashion” goes back to my previous comments on the selectivity in such material. The other big issue is the working capacity of the material proposed here.

A) Selectivity:

Although I 100% agree the proposed process is specific for certain molecules, my question is: Is this being strong enough to counteract for vdW, electrostatic and alternative physi/chemisorption processes. Is the proposed solution able to selectively bind CS₂ in humid conditions?

*The authors argue that “in diamine-appended Mg₂(dobpdc) the presence of high partial pressures of water does not interfere with the chemically specific cooperative adsorption of CO₂”. Again, I agree it will not follow the same “reactive process”, but I will be surprised if the diamine groups are not also hydrophilic. With their results in hand, the fact that “we expect the same trend to hold true to CS₂ adsorption in the presence of water” **is only a hypothesis**. On one hand, a pure water isotherm will let know if the material is hydrophilic or not. On the other hand, a breakthrough curve for CS₂ adsorption in the presence of moisture, reproducing standard operating conditions, will be the definitive answer. CS₂ is toxic, but the authors have shown they can deal with it. If SH₂ produced in contact with water will attack the MOF, then this is another issue not included in the discussion.*

B) Capacity:

*Regarding the previous cartoon for the comparison of pressure: i) “there is no specific industrial process this manuscript is targeting, as such 75 mbar pressure is somewhat arbitrary”. Working at non-optimal conditions, such as 10 mbar makes the bare framework to perform worse than at 75 mbar, but still comparable to the modified one. Why would the authors do that? Again, the comment about the fact that “the location of these condensation steps would be very sensitive to co-adsorbents, making its use unreliable” **is another hypothesis** not demonstrated in the answer. Simply check the optimal conditions for the bare and modified frameworks and compare them.*

My whole point here is that the functionalization has an impact on the pore volume, and this will decrease the working capacity of the material.

In order to be practical and solve this discussion, I ask the authors for mixture isotherms and a full water isotherm. The same safety issues that exist with CS₂ should be valid for humid conditions. In any case, the water isotherms provided are not full isotherms. Saturation pressure of water is 31.7 mbar at 298 K, so please do not stop at 0.15 p/p₀, and make a systematic comparison of CS₂ and water adsorption in both materials, side by side.

In light of statements from the editor, we will not be addressing these comments.